# MoleRanker: Spectrum-Driven Molecular Structure Ranking with Heterogeneous Co-occurrence Graphs

## Abstract

Identifying molecular structures in environmental and biological samples is essential for assessing ecological risks and human health, yet remains highly challenging due to the vast number of unidentified compounds. Tandem mass spectrometry (MS/MS)[1] provides high-throughput spectrum measurements, but existing spectrum-driven identification approaches face key limitations: spectrum-isolated modeling methods are computationally expensive and tend to overlook molecular clustering effects. Moreover, network-based methods typically fail to incorporate environmental co-occurrence across chemical samples, yielding unsatisfactory performance. To address these challenges, we revisit molecular identification as spectrum-driven molecular structure ranking and propose MoleRanker, a novel heterogeneous graph neural network that integrates chemical constraints with environmental co-occurrence patterns. Specifically, we first construct a heterogeneous co-occurrence graph that encodes both *molecular-level chemical clustering effects* and *sample-level environmental co-occurrence correlations*. We then design a multiplex-relation message-passing mechanism to perform information propagation in a relation-aware manner across these heterogeneous relations. We construct four diverse datasets, including in-situ environmental pollutants and human metabolomics, and release them as a benchmark for spectrum-driven molecular structure ranking. Extensive experiments demonstrate that MoleRanker achieves state-of-the-art performance across four datasets. Beyond accuracy, our framework advances both AI methodology and scientific discovery by offering a rigorous benchmark for AI researchers and a practical tool for molecular identification for domain scientists. Code is available at https://anonymous.4open.science/r/MoleRanker.

## 1 Introduction

The ability to discover unidentified molecular structures from environmental and biological samples is critical for monitoring emerging pollutants (Hollender et al., 2017; Zhao et al., 2025), advancing human metabolomics (Gentry et al., 2024; Schuhknecht et al., 2025) and supporting drug development (Meissner et al., 2022; Alarcon-Barrera et al., 2022). High-throughput tandem mass spectrometry (MS/MS) (Gross, 2006) has become the predominant technique for this task, providing fragmentation spectra that encode rich structural information about analytes. However, translating raw spectral data into accurate molecular identification remains a longstanding challenge. The chemical complexity of environmental matrices, the prevalence of unknown compounds, and the lack of reference spectra for many molecules exacerbate the difficulty of reliable structure elucidation.

To address this problem, a variety of computational approaches have been proposed. These approaches can be broadly divided into two categories: spectrum-isolated methods and network-based modeling methods. Traditional spectrum-isolated methods rely on expert knowledge to simulate fragmentation pathways using chemical rules and heuristics (Gerlich & Neumann, 2013; da Silva et al., 2018). Representative in silico fragmentation tools include MetFrag (Ruttkies et al., 2016),

---

[1]Tandem mass spectrometry is abbreviated as MS/MS, where the two MS refer to sequential mass spectrometers, the first for precursor ion selection and the second for fragment ion analysis.

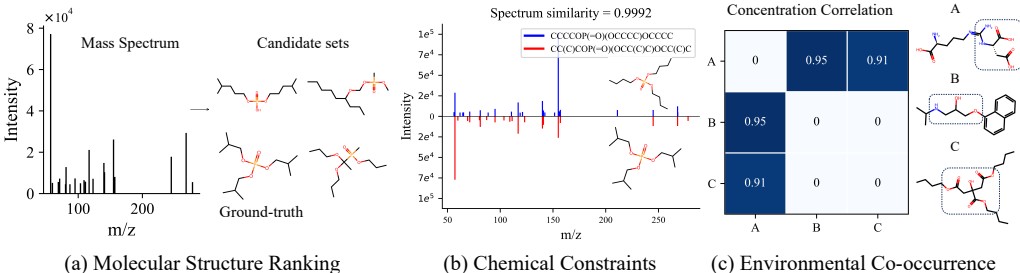

Figure 1: The motivations of our MOLERANKER. (a) *Molecular structure ranking*, where candidate molecules are prioritized according to the mass error tolerance of neutral molecules. (b) *Chemical constraints*: structurally similar compounds often exhibit natural clustering effects, resulting in higher spectral similarity. (c) *Environmental co-occurrence*: compounds sharing functional groups tend to display correlated concentration levels across samples.

CFM-ID (Wang et al., 2021; 2022), and SIRIUS (Dührkop et al., 2019). These methods typically match observed spectra against simulated candidates by comparing peak patterns or predicted fragment ions, but they neglect the broader molecular relationships prevalent in complex environmental and biological samples. Recent spectrum-isolated methods Stravs et al. (2022); Litsa et al. (2023); Goldman et al. (2023); Wang et al. (2025) focus on developing end-to-end deep learning models for de novo molecular generation from MS/MS data. However, these methods require large-scale, high-quality labeled "spectrum–structure" pairs, which limits their applicability, particularly in domains with limited data availability. More recently, molecular networking techniques (Chen et al., 2021; Zhou et al., 2022; Morehouse et al., 2023; Wang et al., 2024) have been developed to exploit spectral similarity through the GNPS platform (Wang et al., 2016). These approaches construct spectrum-based molecular networks to assist in de-replication and label propagation. However, they do not explicitly incorporate environmental co-occurrence patterns across samples, which limits their capacity to capture real-world chemical contexts.

In light of these limitations, we argue that molecular identification should be reformulated as a spectrum-driven molecular structure ranking problem (Figure 1 (a)). In practice, chemists rarely elucidate an unknown compound entirely from scratch. They usually narrow the search space to plausible candidates using prior knowledge of chemical classes, biosynthetic pathways, and sample context, before ranking the most likely structure. This ranking perspective avoids an exhaustive search of the entire chemical space and the complexity of de novo generation. Another motivation is to integrate complementary evidence, namely *chemical constraints* and *environmental co-occurrence*, for more accurate molecular structure ranking. Chemical constraints imply that structurally similar compounds tend to cluster and yield related fragmentation spectra (Figure 1 (b)), while environmental co-occurrence reflects that compounds sharing functional groups tend to display correlated concentration levels across samples (Figure 1 (c)). However, an open challenge remains: *how can chemical constraints and environmental co-occurrence be effectively integrated into a unified model?*

To solve this problem, we propose MOLERANKER, a novel heterogeneous graph neural network that facilitates molecular structure ranking by exploiting the natural inter-molecular interactions. We first construct a heterogeneous molecular co-occurrence graph that simultaneously encodes *molecular-level chemical clustering effects* and *sample-level environmental co-occurrence correlations*. Subsequently, we design a multiplex-relation message-passing mechanism that enables relation-specific information propagation across diverse dependencies. Finally, we introduce a dual-tower scoring strategy for candidate ranking and optimize the model with the Bayesian Personalized Ranking (BPR) objective, mitigating inflated ranking bias and handling the sparsity of positive supervision.

In summary, the main contributions of our work are as follows:

- We revisit molecular identification as a spectrum-driven molecular structure ranking task. In addition, we assemble four datasets to support realistic evaluation, including environmental pollutants collected in situ and human metabolomics from public resources.

- We specify the challenges of spectrum-driven molecular structure ranking and we propose MOL-ERANKER, a heterogeneous graph neural network that integrates information from *chemical constraints* and *environmental co-occurrence*.

- We design a multiplex-relation message-passing mechanism for relation-aware representation fusion, together with a dual-tower scoring strategy and a BPR objective to perform candidate ranking, effectively alleviating inflated ranking bias and the sparsity of positive supervision.

- We conduct comprehensive experiments on the constructed benchmark datasets. Results demonstrate that MOLERANKER achieves state-of-the-art performance, with an average MRR improvement of 12.18%, highlighting its potential for discovering and understanding emerging pollutants and human metabolism.

## 2 RELATED WORKS

**Spectrum-Isolated Molecular Identification.** Tandem mass spectrometry (MS/MS) is a key analytical technique for molecular identification and is widely applied in metabolomics and environmental chemistry. Traditional spectrum-driven methods for molecular identification mainly focus on spectrum-isolated modeling, which can be further divided into two categories. *(1) Spectrum matching.* Rule-based heuristics such as MetFrag (Ruttkies et al., 2016), CFM-ID (Wang et al., 2022), and SIRIUS (Dührkop et al., 2019) match spectra to reference libraries or simulated fragments (Murphy et al., 2023; Young et al., 2024), but they suffer from unsatisfactory performance on novel structures and are computationally expensive. *(2) De novo molecule generation.* Recent methods reconstruct structures directly from spectra without relying on candidate sets (Ludwig et al., 2020). MSNovelist (Stravs et al., 2022) and Spec2Mol (Litsa et al., 2023) adopt sequence-to-sequence architectures. MassGenie (Shrivastava et al., 2021) casts spectrum-to-SMILES as sequence translation with transformers. MIST (Goldman et al., 2023) integrates chemical priors, including formula encoding, neutral-loss features, and substructure prediction, into spectral transformers. MADGEN (Wang et al., 2025) combines scaffold retrieval with spectrum-conditioned generation. DiffMS (Bohde et al., 2025) couples a spectrum-informed transformer encoder with a formula-constrained graph diffusion decoder pre-trained on large-scale pairs of fingerprints and structures. However, these methods require extensive labeled data (Horai et al., 2010; Dührkop et al., 2021; Bushuiev et al., 2024), and to date no spectrum-driven foundation model has achieved broad generalization.

**Network-based Molecular Identification.** Another line of work (Watrous et al., 2012; Bach et al., 2022; Ramos et al., 2019) builds molecular networks by leveraging associations between spectra via the GNPS platform (Wang et al., 2016; Nothias et al., 2020), offering a powerful alternative to spectrum-isolated modeling. NetID (Chen et al., 2021) constructs a global network by linking LC-MS peaks through chemically plausible mass differences and performs joint annotation via global optimization. MetDNA (Shen et al., 2019) uses a reaction network–based recursive strategy, treating MS/MS spectra of seed metabolites as surrogates to annotate structurally related neighbors without requiring large spectral libraries. KGMN (Zhou et al., 2022) integrates a reaction network, MS/MS similarity, and peak correlation layers to propagate annotations from knowns to unknowns at scale. SNAP-MS (Morehouse et al., 2023) aligns molecular sub-networks with chemical families from the Natural Products Atlas to support family-level annotation without reference spectra. APP-ID (Wang et al., 2024) combines machine learning with enhanced fragment matching to annotate diverse PFASs with lower false positives. However, existing network-based annotation methods primarily rely on label propagation over handcrafted networks and overlook molecular co-occurrence patterns arising from shared biological or environmental contexts, which limits their ability to capture sample-level dependencies. In contrast, we incorporate environmental co-occurrence priors into a heterogeneous molecular graph, facilitating more comprehensive information propagation to enhance molecular structure ranking.

## 3 PRELIMINARY

In this section, we first present the problem formulation of molecular structure ranking, and then introduce the benchmark construction of environmental pollutants collected in situ and human metabolomics data from public resources.

## 3.1 PROBLEM FORMULATION

We revisit molecule identification as a spectrum-driven structure ranking problem. Specifically, we define a heterogeneous molecule co-occurrence graph $\mathcal{G}$ as follows:

$$\mathcal{G} = (\mathcal{V}, \mathcal{E}_s, \mathcal{E}_c, X), \tag{1}$$

where $\mathcal{V} = \{v_1, v_2, \ldots, v_N\}$ denotes the set of $N$ nodes, each representing a molecule. The edge set consists of two types. (i) *chemical constraints edges* $\mathcal{E}_s$ encode the spectral similarity between molecules and form the adjacency matrix $\mathbf{A}_s$. (ii) *environmental co-occurrence edges* $\mathcal{E}_c$ capture the concentration correlation of molecules across samples and form the adjacency matrix $\mathbf{A}_c$. Each node $v_i$ is associated with a candidate set

$$\mathcal{C}_i = \{c_{i,1}, c_{i,2}, \ldots, c_{i,M_i}\}, \tag{2}$$

where $M_i$ is the number of candidate molecular structures. Then, the complete candidate set across all $N$ nodes is denoted as

$$\mathcal{C} = \{\mathcal{C}_1, \mathcal{C}_2, \ldots, \mathcal{C}_N\}. \tag{3}$$

The feature matrix is defined as

$$X = [x_1, x_2, \ldots, x_N] \in \mathbb{R}^{N \times d_x}, \quad x_i \in \mathbb{R}^{d_x}, \tag{4}$$

which $x_i$ is the feature vector of node $v_i$, and $d_x$ is the feature dimension. $x_i$ encodes both spectral and domain-specific features. The label set is defined as

$$Y = (y_1, y_2, \ldots, y_N), \quad y_i \in \{0, 1\}^{M_i}, \tag{5}$$

where $y_i$ is a one-hot vector indicating the ground-truth molecular structure in $\mathcal{C}_i$. Therefore, the goal is to learn a molecular structure ranking function $f(\cdot)$ that takes the graph $\mathcal{G}$ and candidate sets $\mathcal{C}$ as input and predicts the ranking scores $\hat{Y}$:

$$f(\mathcal{G}, \mathcal{C}) \rightarrow \hat{Y}. \tag{6}$$

## 3.2 BENCHMARK CONSTRUCTION

We collect and curate MS/MS data covering both environmental pollutants and human metabolites. For the environmental domain, we obtain in-situ sediment samples from Lake Taihu, China, which reflect complex chemical mixtures accumulated over long-term environmental processes. The sediment samples are analyzed by liquid chromatography tandem mass spectrometry (LC-MS/MS), and candidate annotations are validated through spectral library matching and expert curation. For the biomedical domain, we curate publicly available human metabolomics datasets from the Metabo-Lights database (Yurekten et al., 2024). These datasets consist of biological samples such as plasma or urine, annotated with known endogenous compounds. In both domains, we extract MS/MS spectra and associated metadata, including sample-level concentration profiles of each compound. The sample–compound concentration matrix enables us to study environmental co-occurrence patterns across different samples, which we later leverage in our modeling framework. Further details of data acquisition and processing are provided in Appendix D.1.

## 4 METHODOLOGY

## 4.1 OVERVIEW

In this section, we introduce the methodology of spectrum-driven molecular structure ranking and present the detailed architecture of our proposed MOLERANKER. Figure 2 illustrates the overall framework. We begin by constructing a heterogeneous molecular co-occurrence graph from multiple data sources, including MS/MS spectra acquired from experimental samples, compound concentration data across samples, and candidate molecular structures retrieved from the PubChem (Kim et al., 2025) database. We then employ pre-trained encoders to obtain embeddings for spectra and candidates. Subsequently, we perform feature augmentation on each node for semantic enhancement, followed by relation-aware feature fusion to achieve multiplex-relation message-passing. Finally, a dual-tower scoring strategy is applied to predict the ranking results.

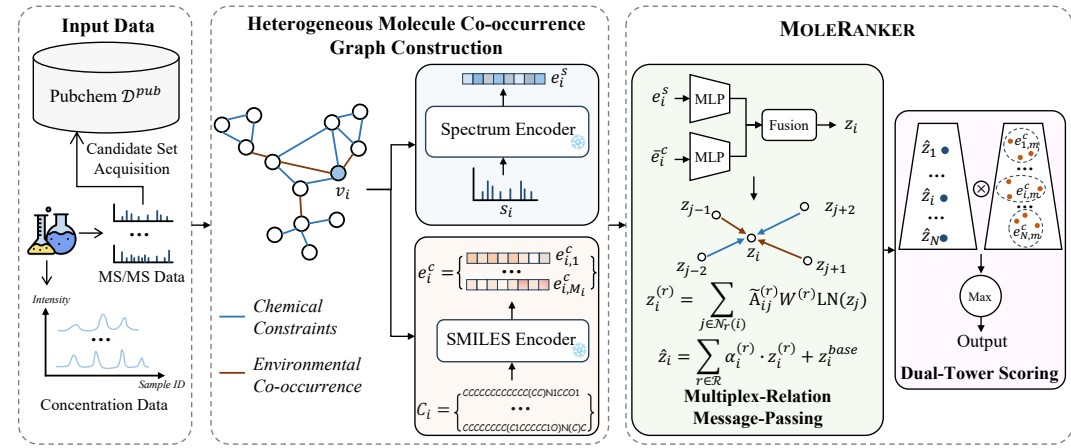

Figure 2: The framework of spectrum-driven molecular structure ranking with our MOLERANKER.

## 4.2 HETEROGENEOUS MOLECULE CO-OCCURRENCE GRAPH CONSTRUCTION

To jointly model spectrum-level and sample-level molecular relationships, we construct a heterogeneous molecular co-occurrence graph $\mathcal{G}$ as follows: (1) retrieving candidate molecules based on precursor mass, (2) encoding both spectra and structures, and (3) connecting nodes using chemical constraints and environmental co-occurrence patterns.

**Candidate Set Acquisition.** Given an MS/MS spectrum with a precursor ion of observed mass $m_i^{ion}$ (computed as the mass-to-charge ratio multiplied by the charge state), we estimate the corresponding neutral molecular mass as $m_i^{neutral} = m_i^{ion} - q_i \cdot m_p$, where $q_i$ is the charge state and $m_p$ is the proton mass. We then retrieve a candidate set of molecules $\mathcal{C}_i = \{c_{i,1}, \ldots, c_{i,M_i}\}$ from the PubChem database $\mathcal{D}^{pub}$ using a mass tolerance filter:

$$\mathcal{C}_i = \left\{ c \in \mathcal{D}^{pub} \mid \left| \mathrm{mass}(c) - m_i^{neutral} \right| \leq \delta \right\}, \tag{7}$$

where $\mathrm{mass}(c)$ denotes the exact neutral mass of candidate $c$, and $\delta$ is a mass tolerance threshold.

**Spectrum Encoder.** Each MS/MS spectrum $s_i$ of node $v_i$ is embedded into $e_i^s \in \mathbb{R}^{d_s}$ via a pretrained spectrum encoder (Bushuiev et al., 2025) $\phi_s(\cdot)$:

$$e_i^s = \phi_s(s_i). \tag{8}$$

**SMILES Encoder.** Each candidate molecule $c_{i,m} \in \mathcal{C}_i$ of node $v_i$ is converted to a molecular descriptor vector $e_{i,m}^c \in \mathbb{R}^{d_{mol}}$ using the ECFP-4 fingerprint function $\phi_c(\cdot)$ implemented in RDKit (Landrum et al., 2025). The complete and the aggregated candidate representation for node $v_i$ are $e_i^c$ and $\bar{e}_i^c$, denoted as follows:

$$e_{i,m}^c = \phi_c(c_{i,m}), \qquad e_i^c = [e_{i,1}^c, \cdots, e_{i,M_i}^c], \qquad \bar{e}_i^c = \frac{1}{M_i} \sum_{m=1}^{M_i} e_{i,m}^c. \tag{9}$$

**Chemical Constraints.** We obtain the chemical constraint graph from the GNPS platform (Wang et al., 2016), which encodes cosine similarity between spectra with optional sparsification:

$$w_{i,j}^{(s)} = \mathrm{GNPS}\left(s_i, s_j\right), \qquad (\mathbf{A}_s)_{i,j} = \mathbb{I}\left[w_{i,j}^{(s)} \geq \tau_s\right] \cdot w_{i,j}^{(s)}, \tag{10}$$

with $\tilde{\mathbf{A}}_s = D_s^{-\frac{1}{2}} \mathbf{A}_s D_s^{-\frac{1}{2}}$ for normalized propagation, where $\mathrm{GNPS}(\cdot)$ is the mapping function in software, $\mathbb{I}$ is identity matrix, $\tau_s$ is a threshold for top-$s$ pruning and $D_s$ is the degree matrix of $\mathbf{A}_s$.

**Environmental Co-occurrence.** Let $r_i \in \mathbb{R}^T$ denote the concentration profile of $v_i$ across $T$ samples. We compute pairwise correlations $\rho(\cdot, \cdot)$ and optionally keep statistically significant edges:

$$w_{i,j}^{(c)} = \rho\left(r_i, r_j\right), \qquad (\mathbf{A}_c)_{i,j} = \mathbb{I}\left[\left|w_{i,j}^{(c)}\right| \geq \tau_c\right] \cdot w_{i,j}^{(c)}, \tag{11}$$

with $\tilde{\mathbf{A}}_c = D_c^{-\frac{1}{2}} \mathbf{A}_c D_c^{-\frac{1}{2}}$ for normalized propagation, where $D_c$ is the degree matrix of $\mathbf{A}_c$.

### 4.3 MULTIPLEX-RELATION MESSAGE-PASSING

First, we concatenate the spectrum projection with the aggregated candidate embedding:

$$z_i = \left[ \mathrm{MLP}(e_i^s) \,\|\, \mathrm{MLP}(\bar{e}_i^c) \right] \in \mathbb{R}^{2d}, \tag{12}$$

where $\mathrm{MLP}(\cdot)$ denotes a multi-layer perceptron, $\|$ denotes concatenation, and $d$ denotes the hidden dimension. Subsequently, we perform a relation-specific graph convolution that incorporates pre-normalization and a residual projection $P_r$ for each relation $r \in \{s, c\}$:

$$z_i^{(r)} = \sum_{j \in \mathcal{N}_r(i)} \tilde{\mathbf{A}}_{ij}^{(r)} \, W^{(r)} \, \mathrm{LN}(z_j), \tag{13}$$

where $\tilde{\mathbf{A}}^{(r)} \in \{\tilde{\mathbf{A}}_s, \tilde{\mathbf{A}}_c\}$, $\mathcal{N}_r(i)$ denotes the set of neighbor nodes for relation $r$ and node $v_i$, $\mathrm{LN}(\cdot)$ denotes LayerNorm. We introduce a lightweight base branch $z_i^{base} = \psi(z_i)$ and fuse relation-specific outputs through attention (Vaswani et al., 2017):

$$\alpha_i^{(r)} = \mathrm{softmax}_r \left( W \left[ z_i^{(r)} \| z_i^{base} \right] \right), \qquad \sum_r \alpha_i^{(r)} = 1, \tag{14}$$

$$\hat{z}_i = \sum_{r \in \{s,c\}} \alpha_i^{(r)} z_i^{(r)} + z_i^{base}, \tag{15}$$

yielding relation-aware node embeddings $\hat{z}_i \in \mathbb{R}^d$ that encode multiplex co-occurrence.

### 4.4 DUAL-TOWER SCORING

We adopt a dual-tower scoring strategy for candidate ranking. The *left tower* produces the node embedding $u_i = \hat{z}_i$ after multiplex propagation, while the *right tower* encodes each candidate $v_{i,m} = e_{i,m}^c$. For each "node-candidate" pair $(i, m)$, we compute a semantic score using temperature-scaled cosine similarity:

$$\hat{u}_i = \frac{U u_i}{\|U u_i\|_2}, \qquad \hat{v}_{i,m} = \frac{V v_{i,m}}{\|V v_{i,m}\|_2}, \qquad S_{i,m} = \frac{\langle \hat{u}_i, \hat{v}_{i,m} \rangle}{\tau}, \tag{16}$$

where $\tau > 0$ is a learnable temperature, and $U, V$ are trainable parameters. The final ranking result for node $v_i$ is obtained by sorting $\{S_{i,m}\}_{m=1}^{M_i}$.

### 4.5 OPTIMIZATION ALGORITHM

To effectively cope with the large candidate set, we optimize MOLERANKER with a pairwise Bayesian Personalized Ranking (BPR) objective:

$$\mathcal{L}_{\mathrm{BPR}} = -\frac{1}{|\mathcal{D}|} \sum_i \mathbb{E}_{m^+ \in \mathcal{P}_i} \, \mathbb{E}_{m^- \sim \mathcal{N}_i} \left[ \log \sigma \left( S_{i,m^+} - S_{i,m^-} \right) \right], \tag{17}$$

where $\sigma$ is sigmoid function, $\mathcal{P}_i$ and $\mathcal{N}_i$ are the sets of positive and negative candidates, respectively. This objective directly enforces positive candidates to be ranked higher than negatives. We conclude the complete algorithm in Appendix C.

## 5 EXPERIMENTS

In this section, we evaluate the effectiveness of our proposed MOLERANKER for molecular structure ranking on both environmental pollutants and human metabolomics datasets. The comprehensive analysis is presented with the aim of answering the following research questions (RQs): **(RQ1)** Can MOLERANKER effectively solve the molecular structure ranking task, and how does its performance compare with baselines? **(RQ2)** What is the role of the heterogeneous molecular co-occurrence graph in MOLERANKER, and how does it enhance the model's ability to rank candidate molecules? **(RQ3)** What information does the heterogeneous molecular co-occurrence graph

contribute to molecular structure ranking? **(RQ4)** How do the graph construction thresholds and key hyperparameters of model influence the performance of MOLERANKER? In addition, how does MOLERANKER perform under other pre-trained SMILES encoders? **(RQ5)** How does the runtime efficiency of MOLERANKER compare with the baselines?

## 5.1 EXPERIMENT SETUP

**Datasets.** To evaluate MOLERANKER, we use four datasets of distinct origins. *Sediments* consists of environmental pollutants collected in situ, reflecting real-world environmental complexity. In contrast, *MTBLS146* (Luan et al., 2014), *MTBLS265* (Chaleckis et al., 2016), and *MTBLS746* (Geier et al., 2020) are human metabolomics datasets curated from public databases, providing complementary biomedical perspectives. Further details of the datasets are provided in Appendix D.1.

**Baselines.** We compare MoleRanker with three categories of baselines. The first category includes expert knowledge–dominated methods: MetFrag (Ruttkies et al., 2016), CFM-ID (Wang et al., 2022), and SIRIUS (Dührkop et al., 2019), which rely on curated chemical rules and fragmentation knowledge for molecular identification. The second category consists of spectrum-isolated learning approaches and homogeneous network-based methods, including Random Forest (RF) (Rigatti, 2017), XGBoost (Chen & Guestrin, 2016), MLP, GCN (Kipf & Welling, 2017), GAT (Veličković et al., 2018), and GraphSAGE (Hamilton et al., 2017), which are data-driven but do not incorporate multiplex molecular co-occurrence. In addition, the third category consists of heterogeneous graph neural networks that we adapt to the molecular structure ranking task, including R-GCN (Schlichtkrull et al., 2018), HAN (Wang et al., 2019), HetGNN (Zhang et al., 2019), HGT (Hu et al., 2020), and SeHGNN (Yang et al., 2023). More details about baselines are presented in Appendix D.2. We additionally include results using *de novo* generation methods as baselines, which are reported in the Appendix E.

**Metrics.** To assess ranking performance, we adopt two widely used metrics: Hits@K and Mean Reciprocal Rank (MRR). Further details are provided in Appendix D.3.

**Implementation.** We split the nodes of each dataset into training, validation, and testing sets following a 7:1:2 ratio. All experiments are repeated 5 times with different seeds, and we report the results as the mean ± standard deviation. Additional implementation details are provided in Appendix D.4.

## 5.2 OVERALL PERFORMANCE

To address RQ1, we evaluate MOLERANKER against all baselines on the spectrum-driven molecular structure ranking task, with results reported in Table 1.

❶ MOLERANKER *achieves state-of-the-art performance across all four datasets.* Concretely, MOLERANKER improves Hits@1, Hits@10, Hits@20, and MRR by **7.91%**, **8.13%**, **8.42%**, and **12.18%** on average compared with the strongest baselines from the first two categories, respectively, demonstrating consistent improvements in both Hits@K accuracy and overall ranking quality. ❷ MOLERANKER *substantially outperforms expert knowledge–dominated methods on human metabolomics datasets.* Methods such as MetFrag, CFM-ID, and SIRIUS simulate fragmentation via chemical rules and rely on manually crafted scoring schemes, which limits generalizability to unseen chemical spaces. In practice, these heuristics often assign similar scores to multiple candidates, leading to frequent ties and weakened ranking capability, whereas MOLERANKER maintains discriminative scores. ❸ MOLERANKER *surpasses spectrum-isolated machine learning models and homogeneous GNNs.* By explicitly modeling *chemical constraints* and *environmental co-occurrence*, MOLERANKER leverages a multiplex relation message-passing mechanism to integrate complementary evidence, capturing the molecular-level chemical clustering effects and sample-level environmental co-occurrence correlations. This relation-aware propagation yields more informative node representations and more accurate candidate rankings than spectrum-isolated classifiers and homogeneous GNNs. Furthermore, MoleRanker continues to exhibit superior performance compared with other heterogeneous GNN models, indicating that our model architecture is better suited for capturing chemical constraints and environmental co-occurrence among molecules.

Table 1: **MOLERANKER achieves the state-of-the-art performance across all datasets**. We run the experiments 5 times and show the mean ± standard deviation of the results. Hits@K values are reported in percentage, and MRR is reported as a raw score. The best results are highlighted in **bold**, and the second-best results are underlined.

| Methods | Sediments | | | | MTBLS146 | | | |
|---|---|---|---|---|---|---|---|---|
| | Hits@1↑ | Hits@10↑ | Hits@20↑ | MRR↑ | Hits@1↑ | Hits@10↑ | Hits@20↑ | MRR↑ |
| MetFrag | 9.30 | 23.26 | 30.23 | 0.144 | 3.70 | 3.70 | 3.70 | 0.037 |
| CFM-ID | 6.98 | 16.28 | 25.58 | 0.116 | 0.00 | 0.00 | 0.00 | 0.000 |
| SIRIUS | 16.28 | 20.93 | 23.26 | 0.182 | 0.00 | 0.00 | 0.00 | 0.000 |
| RF | $8.37_{\pm1.14}$ | $19.53_{\pm1.14}$ | $21.86_{\pm1.14}$ | $0.127_{\pm0.007}$ | $20.00_{\pm5.54}$ | $34.07_{\pm3.63}$ | $46.67_{\pm3.78}$ | $0.248_{\pm0.039}$ |
| XGBoost | $6.98_{\pm0.00}$ | $25.58_{\pm0.00}$ | $32.56_{\pm0.00}$ | $0.132_{\pm0.000}$ | $37.04_{\pm0.00}$ | $70.37_{\pm0.00}$ | $70.37_{\pm0.00}$ | $0.480_{\pm0.000}$ |
| MLP | $7.44_{\pm0.93}$ | $26.51_{\pm3.15}$ | $39.07_{\pm2.71}$ | $0.141_{\pm0.004}$ | $52.59_{\pm1.48}$ | $66.67_{\pm3.31}$ | $68.15_{\pm2.96}$ | $0.563_{\pm0.020}$ |
| GCN | $8.37_{\pm1.86}$ | $23.26_{\pm2.08}$ | $36.28_{\pm1.86}$ | $0.136_{\pm0.014}$ | $51.85_{\pm4.06}$ | $71.11_{\pm4.32}$ | $74.07_{\pm2.34}$ | $0.574_{\pm0.024}$ |
| GAT | $8.37_{\pm1.86}$ | $21.40_{\pm1.74}$ | $34.42_{\pm1.74}$ | $0.136_{\pm0.011}$ | $49.63_{\pm7.63}$ | $71.11_{\pm5.93}$ | $72.59_{\pm2.96}$ | $0.555_{\pm0.068}$ |
| GraphSAGE | $7.44_{\pm2.28}$ | $20.47_{\pm0.93}$ | $33.49_{\pm3.15}$ | $0.127_{\pm0.017}$ | $51.85_{\pm2.34}$ | $72.59_{\pm1.81}$ | $75.56_{\pm2.96}$ | $0.579_{\pm0.013}$ |
| R-GCN | $11.63_{\pm1.47}$ | $35.81_{\pm2.37}$ | $45.12_{\pm1.86}$ | $0.195_{\pm0.010}$ | $53.33_{\pm5.54}$ | $70.37_{\pm4.06}$ | $73.33_{\pm4.32}$ | $0.577_{\pm0.050}$ |
| HAN | $11.16_{\pm0.93}$ | $33.02_{\pm1.43}$ | $43.26_{\pm2.37}$ | $0.186_{\pm0.009}$ | $52.59_{\pm4.32}$ | $68.15_{\pm3.78}$ | $72.59_{\pm1.81}$ | $0.568_{\pm0.034}$ |
| HetGNN | $12.56_{\pm2.37}$ | $36.28_{\pm2.37}$ | $44.19_{\pm2.94}$ | $0.199_{\pm0.020}$ | $48.89_{\pm3.63}$ | $68.89_{\pm3.78}$ | $72.59_{\pm5.02}$ | $0.558_{\pm0.025}$ |
| HGT | $12.56_{\pm2.37}$ | $34.88_{\pm0.00}$ | $43.26_{\pm2.37}$ | $0.197_{\pm0.016}$ | $50.37_{\pm3.78}$ | $72.59_{\pm1.81}$ | $74.07_{\pm2.34}$ | $0.563_{\pm0.026}$ |
| SeHGNN | $11.16_{\pm0.93}$ | $\mathbf{37.21_{\pm4.88}}$ | $44.65_{\pm2.71}$ | $0.193_{\pm0.011}$ | $51.85_{\pm4.06}$ | $70.37_{\pm3.31}$ | $74.07_{\pm2.34}$ | $0.570_{\pm0.032}$ |
| MOLERANKER | $\mathbf{16.74_{\pm0.93}}$ | $33.02_{\pm2.71}$ | $\mathbf{48.84_{\pm2.08}}$ | $\mathbf{0.226_{\pm0.005}}$ | $\mathbf{57.04_{\pm1.81}}$ | $\mathbf{73.33_{\pm2.77}}$ | $\mathbf{77.04_{\pm1.48}}$ | $\mathbf{0.604_{\pm0.021}}$ |

| Methods | MTBLS265 | | | | MTBLS746 | | | |
|---|---|---|---|---|---|---|---|---|
| | Hits@1↑ | Hits@10↑ | Hits@20↑ | MRR↑ | Hits@1↑ | Hits@10↑ | Hits@20↑ | MRR↑ |
| MetFrag | 0.00 | 0.00 | 0.00 | 0.000 | 0.00 | 0.00 | 0.00 | 0.001 |
| CFM-ID | 0.00 | 0.00 | 0.00 | 0.000 | 0.00 | 0.00 | 0.00 | 0.000 |
| SIRIUS | 0.00 | 0.00 | 0.00 | 0.000 | 0.00 | 0.00 | 0.00 | 0.000 |
| RF | $6.67_{\pm1.69}$ | $9.60_{\pm1.00}$ | $13.60_{\pm2.29}$ | $0.080_{\pm0.016}$ | $1.86_{\pm1.74}$ | $8.84_{\pm4.00}$ | $16.28_{\pm3.29}$ | $0.045_{\pm0.016}$ |
| XGBoost | $10.67_{\pm0.00}$ | $22.67_{\pm0.00}$ | $\mathbf{30.67_{\pm0.00}}$ | $0.146_{\pm0.000}$ | $2.33_{\pm0.00}$ | $9.30_{\pm0.00}$ | $23.26_{\pm0.00}$ | $0.050_{\pm0.000}$ |
| MLP | $6.67_{\pm0.84}$ | $18.67_{\pm1.46}$ | $24.80_{\pm1.36}$ | $0.111_{\pm0.006}$ | $4.19_{\pm0.93}$ | $15.35_{\pm2.37}$ | $22.79_{\pm0.93}$ | $0.072_{\pm0.005}$ |
| GCN | $6.40_{\pm0.53}$ | $17.87_{\pm2.99}$ | $24.53_{\pm4.51}$ | $0.110_{\pm0.012}$ | $4.07_{\pm1.93}$ | $18.60_{\pm2.33}$ | $23.26_{\pm0.00}$ | $0.084_{\pm0.015}$ |
| GAT | $7.47_{\pm0.65}$ | $19.20_{\pm2.00}$ | $26.13_{\pm1.60}$ | $0.120_{\pm0.006}$ | $4.19_{\pm2.71}$ | $14.42_{\pm2.71}$ | $20.93_{\pm3.60}$ | $0.078_{\pm0.023}$ |
| GraphSAGE | $6.13_{\pm0.65}$ | $17.87_{\pm2.47}$ | $24.53_{\pm1.36}$ | $0.108_{\pm0.011}$ | $4.19_{\pm0.93}$ | $14.88_{\pm1.86}$ | $21.86_{\pm2.79}$ | $0.078_{\pm0.005}$ |
| R-GCN | $8.80_{\pm1.60}$ | $22.67_{\pm1.46}$ | $29.07_{\pm2.13}$ | $0.142_{\pm0.013}$ | $3.26_{\pm1.14}$ | $19.53_{\pm1.14}$ | $25.58_{\pm0.00}$ | $0.084_{\pm0.003}$ |
| HAN | $7.73_{\pm1.00}$ | $22.67_{\pm1.89}$ | $29.07_{\pm2.13}$ | $0.130_{\pm0.006}$ | $4.19_{\pm0.93}$ | $19.53_{\pm1.14}$ | $25.12_{\pm0.93}$ | $0.090_{\pm0.007}$ |
| HetGNN | $6.13_{\pm0.65}$ | $21.07_{\pm2.29}$ | $26.40_{\pm3.71}$ | $0.114_{\pm0.006}$ | $\mathbf{4.65_{\pm0.00}}$ | $19.07_{\pm1.74}$ | $25.58_{\pm0.00}$ | $0.094_{\pm0.009}$ |
| HGT | $8.00_{\pm0.84}$ | $22.67_{\pm1.46}$ | $26.93_{\pm2.44}$ | $0.130_{\pm0.009}$ | $4.19_{\pm0.93}$ | $\mathbf{20.47_{\pm0.93}}$ | $25.58_{\pm0.00}$ | $0.091_{\pm0.005}$ |
| SeHGNN | $9.60_{\pm1.00}$ | $21.87_{\pm2.75}$ | $27.47_{\pm2.17}$ | $0.140_{\pm0.015}$ | $3.26_{\pm1.14}$ | $20.00_{\pm2.37}$ | $25.58_{\pm0.00}$ | $0.088_{\pm0.006}$ |
| MOLERANKER | $\mathbf{11.67_{\pm2.19}}$ | $\mathbf{23.67_{\pm2.38}}$ | $29.67_{\pm1.45}$ | $\mathbf{0.157_{\pm0.013}}$ | $\mathbf{4.65_{\pm0.00}}$ | $19.07_{\pm1.74}$ | $25.58_{\pm0.00}$ | $\mathbf{0.095_{\pm0.008}}$ |

Table 2: Ablation study of MOLERANKER on four datasets.

| Methods | Sediments | | MTBLS146 | | MTBLS265 | | MTBLS746 | |
|---|---|---|---|---|---|---|---|---|
| | Hits@1↑ | MRR↑ | Hits@1↑ | MRR↑ | Hits@1↑ | MRR↑ | Hits@1↑ | MRR↑ |
| w/o graph | $9.30_{\pm1.47}$ | $0.176_{\pm0.009}$ | $51.85_{\pm7.03}$ | $0.580_{\pm0.039}$ | $8.00_{\pm1.89}$ | $0.133_{\pm0.015}$ | $3.26_{\pm1.14}$ | $0.080_{\pm0.009}$ |
| w/o $\mathbf{A}_s$ | $12.56_{\pm2.79}$ | $0.197_{\pm0.019}$ | $54.81_{\pm5.93}$ | $0.595_{\pm0.036}$ | $8.53_{\pm2.47}$ | $0.136_{\pm0.018}$ | $3.72_{\pm1.14}$ | $0.082_{\pm0.016}$ |
| w/o $\mathbf{A}_c$ | $10.23_{\pm2.37}$ | $0.186_{\pm0.017}$ | $55.56_{\pm4.81}$ | $0.601_{\pm0.028}$ | $8.53_{\pm1.36}$ | $0.135_{\pm0.013}$ | $3.72_{\pm1.14}$ | $0.092_{\pm0.006}$ |
| w/ BCE | $12.09_{\pm0.93}$ | $0.196_{\pm0.016}$ | $37.78_{\pm1.48}$ | $0.440_{\pm0.013}$ | $4.53_{\pm0.65}$ | $0.077_{\pm0.004}$ | $0.00_{\pm0.00}$ | $0.018_{\pm0.003}$ |
| w/ InfoNCE | $2.79_{\pm2.71}$ | $0.055_{\pm0.036}$ | $7.41_{\pm7.77}$ | $0.121_{\pm0.092}$ | $1.60_{\pm2.59}$ | $0.030_{\pm0.033}$ | $0.93_{\pm1.86}$ | $0.036_{\pm0.024}$ |
| MOLERANKER | $\mathbf{16.74_{\pm0.93}}$ | $\mathbf{0.226_{\pm0.005}}$ | $\mathbf{57.04_{\pm1.81}}$ | $\mathbf{0.604_{\pm0.021}}$ | $\mathbf{11.67_{\pm2.19}}$ | $\mathbf{0.157_{\pm0.013}}$ | $\mathbf{4.65_{\pm0.00}}$ | $\mathbf{0.095_{\pm0.008}}$ |

## 5.3 ABLATION STUDY OF ARCHITECTURES AND RANKING LOSS

To address RQ2, we conduct ablations along two dimensions: (i) *architecture*, by selectively removing relations from the heterogeneous graph, and (ii) *ranking loss*, by substituting the pairwise BPR objective with pointwise or contrastive alternatives. For architecture, we evaluate **w/o $\mathbf{A}_s$** (removing chemical constraints), **w/o $\mathbf{A}_c$** (removing environmental co-occurrence), and **w/o graph** (removing both, i.e., spectrum-isolated). For ranking loss, we compare **w/ BCE** (binary cross-entropy) and **w/ InfoNCE** against our default **BPR**. Results are summarized in Table 2.

❶ *Both chemical constraints and environmental co-occurrence positively contribute to* MOLER-ANKER. Removing either relation (**w/o $\mathbf{A}_s$** or **w/o $\mathbf{A}_c$**) consistently degrades performance across

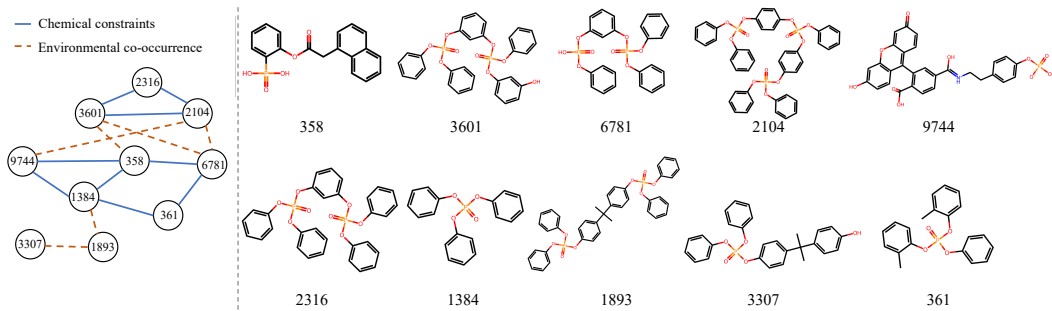

Figure 3: **Chemical constraints and environmental co-occurrence reveal natural molecular clustering patterns, facilitating accurate structure ranking.** (*Left*) Heterogeneous graph from a Sediments subset. (*Right*) MOLERANKER predictions on selected Sediments compounds reveal similar organophosphate ester structures.

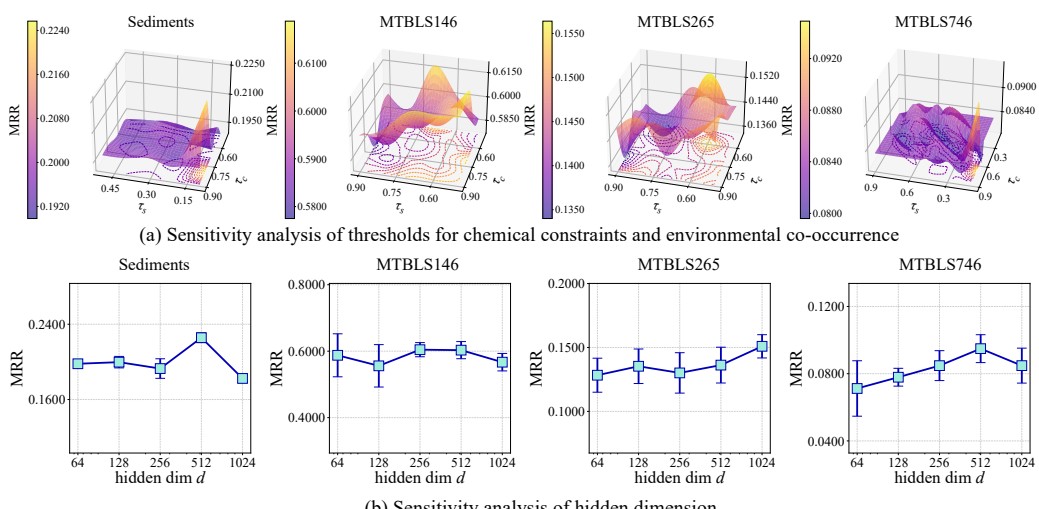

(a) Sensitivity analysis of thresholds for chemical constraints and environmental co-occurrence

(b) Sensitivity analysis of hidden dimension

Figure 4: Sensitivity analysis of MOLERANKER.

datasets, and **w/o graph** yields the largest drop, confirming the complementary nature of the two relations. Intuitively, structurally similar compounds exhibit spectrum-level aggregation and higher spectral similarity, while environmental co-occurrence provides additional guidance: compounds sharing functional groups tend to co-vary in concentration across samples. Our multiplex-relation message passing leverages both signals, leading to more discriminative node representations and improved ranking. ❷ *BPR is better aligned with spectrum-driven ranking than BCE and InfoNCE.* In spectrum-driven molecular structure ranking, each node faces a large candidate set with extreme class imbalance, i.e., few positives and many negatives. The pairwise BPR objective directly optimizes relative ordering, which better matches the molecular structure ranking task. In contrast, BCE is pointwise, treating positives/negatives independently and thus being sensitive to class imbalance, while InfoNCE focuses on representation alignment within a temperature-scaled softmax, which does not directly optimize the pairwise rank gaps.

## 5.4 EFFECTIVE ANALYSIS OF HETEROGENEOUS MOLECULAR CO-OCCURRENCE GRAPH

To answer RQ3, we select a representative subgraph from the Sediments dataset and visualize the prediction results of the compounds in this subgraph. As shown in Figure 3, chemical constraints and environmental co-occurrence jointly induce a natural clustering effect among the selected nodes in the Sediments dataset. Furthermore, the top-1 predictions of MOLERANKER reveal that the identified structures are highly similar and belong to the class of organophosphate esters. These findings demonstrate that *chemical constraints and environmental co-occurrence jointly capture natural molecular clustering and sample-level co-occurrence patterns*, thereby facilitating more accurate molecular structure ranking.

Table 3: Performance of MOLERANKER with different SMILES encoders (DreaMS as spectral encoder). We report mean $\pm$ standard deviation over 5 runs. Hits@K are in %, MRR is a raw score.

| Methods | Sediments | | | | MTBLS146 | | | |
|---|---|---|---|---|---|---|---|---|
| | Hits@1↑ | Hits@10↑ | Hits@20↑ | MRR↑ | Hits@1↑ | Hits@10↑ | Hits@20↑ | MRR↑ |
| MOLERANKER(DreaMS+RDKit) | $16.74_{\pm0.93}$ | $33.02_{\pm2.71}$ | $48.84_{\pm2.08}$ | $0.226_{\pm0.005}$ | $57.04_{\pm1.81}$ | $73.33_{\pm2.77}$ | $77.04_{\pm1.48}$ | $0.604_{\pm0.021}$ |
| MOLERANKER(DreaMS+MolT5) | $27.91_{\pm1.47}$ | $57.67_{\pm6.31}$ | $69.30_{\pm4.26}$ | $0.365_{\pm0.021}$ | $60.74_{\pm10.63}$ | $86.67_{\pm5.02}$ | $90.37_{\pm4.44}$ | $0.696_{\pm0.089}$ |
| MOLERANKER(DreaMS+MolFormer) | $30.23_{\pm0.00}$ | $59.53_{\pm1.86}$ | $65.58_{\pm2.28}$ | $0.397_{\pm0.004}$ | $57.78_{\pm5.02}$ | $71.85_{\pm5.02}$ | $80.00_{\pm2.96}$ | $0.631_{\pm0.045}$ |
| Methods | MTBLS265 | | | | MTBLS746 | | | |
| | Hits@1↑ | Hits@10↑ | Hits@20↑ | MRR↑ | Hits@1↑ | Hits@10↑ | Hits@20↑ | MRR↑ |
| MOLERANKER(DreaMS+RDKit) | $11.67_{\pm2.19}$ | $23.67_{\pm2.38}$ | $29.67_{\pm1.45}$ | $0.157_{\pm0.013}$ | $4.65_{\pm0.00}$ | $19.07_{\pm1.74}$ | $25.58_{\pm0.00}$ | $0.095_{\pm0.008}$ |
| MOLERANKER(DreaMS+MolT5) | $85.33_{\pm1.46}$ | $93.87_{\pm1.60}$ | $95.47_{\pm1.81}$ | $0.883_{\pm0.014}$ | $67.44_{\pm2.94}$ | $81.86_{\pm4.74}$ | $85.58_{\pm2.28}$ | $0.731_{\pm0.020}$ |
| MOLERANKER(DreaMS+MolFormer) | $75.47_{\pm1.81}$ | $82.93_{\pm0.53}$ | $84.53_{\pm1.07}$ | $0.783_{\pm0.011}$ | $31.63_{\pm3.15}$ | $52.09_{\pm2.37}$ | $63.26_{\pm4.97}$ | $0.396_{\pm0.028}$ |

## 5.5 HYPERPARAMETER SENSITIVITY AND MOLECULAR REPRESENTATION ANALYSIS

To answer RQ4, we examine how graph construction thresholds and hidden dimension affect MOLERANKER. As shown in Figure 4 (a), *performance remains stable across a reasonable range of sparsification thresholds $\tau_s$ and $\tau_c$*, indicating that the heterogeneous graph is relative robust. Moreover, Figure 4 (b) shows that compressing the original 2048-dimensional embeddings to 512 yields consistently better results by reducing redundancy and highlighting informative features. Thus, *a moderate hidden dimension offers the trade-off between compactness and information preservation.*

We further evaluate MoleRanker with MolT5 (Edwards et al., 2022) and MolFormer (Ross et al., 2022) as alternative SMILES encoders to examine whether the framework remains effective under different molecular representations. Since MolT5 and MolFormer are pretrained on large-scale SMILES corpora and thus capture richer chemical semantics than traditional RDKit ECFP-4 fingerprints, MoleRanker equipped with these pretrained encoders achieves notably stronger performance. This confirms that our framework can effectively leverage improved molecular representations.

## 5.6 RUNTIME EFFICIENCY ANALYSIS

To answer RQ5, we provide a comprehensive comparison of computational efficiency across all methods. Specifically, we report the inference runtime of MOLERANKER alongside traditional molecular identification tools (MetFrag, CFM-ID, and SIRIUS) on all four datasets, and MOLERANKER is consistently

Table 4: Comparison of Inference Runtime.

| Methods | Sediments | MTBLS146 | MTBLS265 | MTBLS746 |
|---|---|---|---|---|
| MetFrag | 2.5h | 3min | 1.2h | 16min |
| CFM-ID | 36h | 109h | 89h | 99h |
| SIRIUS | 2min | 3min | 3h14min | 3min |
| MoleRanker | 0.6s | 0.6s | 1s | 0.7s |

and substantially faster. We further include per-iteration training time and inference time for MOLERANKER and the machine learning baselines in Appendix F, showing that their computational costs are comparable. Overall, MOLERANKER achieves both high accuracy and excellent efficiency.

## 6 CONCLUSION

In this paper, we propose MOLERANKER, a heterogeneous graph neural network designed to address the underexplored challenge of molecular structure ranking. We construct a heterogeneous molecular co-occurrence graph and employ a multiplex-relation message-passing mechanism with a dual-tower scoring strategy for discriminative and accurate ranking. We conduct extensive experiments on four datasets and the results demonstrate that MOLERANKER achieves state-of-the-art performance across environmental pollutants and human metabolomics. Additionally, MOLERANKER provides a new perspective on spectrum-driven molecular structure ranking, advancing the integration of spectral, structural, and environmental signals within a unified heterogeneous graph framework. In the future, we aim to combine multimodal data sources to build more expressive heterogeneous molecular co-occurrence networks, thereby further improving the applicability and robustness of MOLERANKER.

## 7 ETHICS STATEMENT

This work complies with the ICLR Code of Ethics. It does not involve human subjects, sensitive personal data, or experiments that could cause harm to individuals, communities, or the environment. The proposed methods are intended solely for general machine learning research and present no foreseeable risks of misuse or harmful applications. To the best of our knowledge, this study raises no conflicts of interest or ethical concerns.

## 8 REPRODUCIBILITY STATEMENT

We have taken concrete steps to ensure the reproducibility of our work. Full implementation details of the models, experimental setup are provided in Appendix C and D. To further support reproducibility, we release our code in an anonymous public repository at https://anonymous.4open.science/r/MoleRanker . These resources enable independent verification of our results.

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

## A NOTATIONS

In Table 5, we list the main symbols and notations used throughout the paper.

Table 5: Important notations and corresponding descriptions.

| Symbols | Definition |
|---|---|
| $\mathcal{G} = (\mathcal{V}, \mathcal{E}_s, \mathcal{E}_c, X)$ | Heterogeneous molecule co-occurrence graph |
| $\mathcal{V} = \{v_1, \ldots, v_N\}$ | Set of $N$ molecular nodes |
| $\mathcal{E}_s, \mathcal{E}_c$ | Chemical constraints/ environmental co-occurrence edge sets |
| $\mathbf{A}_s, \mathbf{A}_c$ | Adjacency matrices of $\mathcal{E}_s$ and $\mathcal{E}_c$ |
| $\tilde{\mathbf{A}}_s, \tilde{\mathbf{A}}_c$ | Normalized adjacency matrices for message passing |
| $X = [x_1, \ldots, x_N]$ | Input feature matrix of nodes |
| $x_i \in \mathbb{R}^{d_x}$ | Feature vector of node $v_i$ |
| $\mathcal{C}_i = \{c_{i,1}, \ldots, c_{i,M_i}\}$ | Candidate set of node $v_i$, consisting of $M_i$ molecular structures |
| $\mathcal{C} = \{\mathcal{C}_1, \ldots, \mathcal{C}_N\}$ | All candidate sets of molecular nodes |
| $Y = (y_1, \ldots, y_N)$ | Ground-truth label vectors of $N$ nodes |
| $\hat{Y}$ | Predicted ranking results of $N$ nodes |
| $y_i \in \{0, 1\}^{M_i}$ | Ground-truth label vectors of node $v_i$ |
| $f(\cdot)$ | molecular structure ranking function |
| $\mathcal{D}^{pub}$ | Pubchem database |
| $m_i^{ion}$ | Precursor ion mass |
| $m_i^{neural}$ | Neutral molecular mass |
| $\delta$ | Mass tolerance threshold for candidate set acquisition |
| $s_i$ | MS/MS spectrum of node $v_i$, serving as one of the input features in $x_i$ |
| $e_i^s \in \mathbb{R}^{d_s}$ | Spectrum embedding of node $v_i$ |
| $e_{i,m}^c \in \mathbb{R}^{d_{\text{mol}}}$ | SMILES embedding of candidate $c_{i,m}$ |
| $e_i^c$ | SMILES embedding of candidate sets $C_i$ |
| $\bar{e}_i^c$ | Mean embedding over candidate set $\mathcal{C}_i$ |
| $w_{i,j}^{(s)}, w_{i,j}^{(c)}$ | Edge weight of $\mathbf{A}_s$ and $\mathbf{A}_c$ |
| $\tau_s, \tau_c$ | Threshold of chemical and environmental co-occurrence |
| $\mathbb{I}$ | Identity matrix |
| $D_s, D_c$ | Degree matrix of $\mathbf{A}_s$ and $\mathbf{A}_c$ |
| $\|$ | Concatenation operation |
| $z_i \in \mathbb{R}^{2d}$ | Augmented node feature vector after MLP, $d$ is the hidden dimension |
| $z_i^{(s)}, z_i^{(c)}$ | Relation-specific node features |
| $\mathcal{N}_r(i)$ | The set of neighbor nodes for relation $r$ and node $v_i$ |
| $z_i^{base}$ | Base branch output for attention fusion |
| $\alpha_i^{(r)}$ | Attention weight for relation $r \in \{s, c\}$ |
| $u_i = z_i$ | Node embedding after multiplex message passing |
| $v_{i,m} = e_{i,m}^c$ | Candidate embedding |
| $\hat{u}_i, \hat{v}_{i,m}$ | Normalized final embeddings for ranking |
| $S_{i,m}$ | Final ranking score of candidate $c_{i,m}$ for node $v_i$ |
| $\tau$ | Learnable temperature parameter |
| $\mathcal{L}_{\text{BPR}}$ | Pairwise ranking loss function |
| $\mathcal{P}_i, \mathcal{N}_i$ | Positive / negative candidate sets of node $v_i$ |
| $\phi_s(\cdot), \phi_c(\cdot)$ | Spectrum encoder / molecular descriptor encoder |
| $\psi(\cdot)$ | Residual base MLP branch for fusion |
| $\sigma(\cdot)$ | Sigmoid function |
| $U, V$ | Projection matrices for final ranking embedding |
| $W, W^{(r)}$ | Learnable weight matrices |
| $T$ | Number of samples (used in concentration profiles) |
| $r_i \in \mathbb{R}^T$ | Concentration profile of node $v_i$ across samples |

## B  BACKGROUND ON SPECIALIZED TERMS

In this section, we provide background details on the following specialized terms:

- **MS/MS** (Gross, 2006): Tandem mass spectrometry (MS/MS) is an analytical technique in which molecules are first ionized and selected as precursor ions in one mass spectrometer, then fragmented and analyzed in a second stage, providing fragmentation spectra that encode structural information about the molecules.

- **SMILES** (Lipkowitz & Boyd, 1996): The Simplified Molecular Input Line Entry System (SMILES) is a text-based notation that encodes the structure of a chemical compound as a string, representing atoms, bonds, and connectivity in a compact and machine-readable form. Common variants include *Canonical SMILES*, which ensures a unique and deterministic representation of a molecule, *Isomeric SMILES*, which incorporates stereochemistry and isotopic information.

- **PubChem** (Kim et al., 2025): PubChem is a publicly available chemical database maintained by the U.S. National Institutes of Health (NIH), providing standardized information on chemical structures, properties, bioactivities, and associated experimental data or literature. It currently hosts information on more than 100 million chemical compounds and can be accessed at https://pubchem.ncbi.nlm.nih.gov/.

- **GNPS** Wang et al. (2016): The Global Natural Products Social Molecular Networking (GNPS) platform is an open-access knowledge base for mass spectrometry data that enables the sharing, visualization, and molecular networking of MS/MS spectra, thereby supporting community-driven compound annotation and discovery. It is publicly available at https://gnps.ucsd.edu/.

- **Precursor Mass** (Gowda & Djukovic, 2014): The mass of the precursor (or parent) ion selected for fragmentation in MS/MS experiments. It is obtained by converting the measured $m/z$ value of the ion into its neutral molecular mass, taking into account the ion's charge state and the proton mass.

- **Neutral Molecular Mass** (Gowda & Djukovic, 2014): The neutral molecular mass represents the actual molecular mass of a compound without any added or removed protons or charges. It can be calculated from the precursor $m/z$ and the ion's charge state by correcting for the proton mass, and it serves as the basis for retrieving candidate molecules from chemical databases such as PubChem.

## C  OPTIMIZATION ALGORITHM OF MOLERANKER

In this section, we introduce the optimization procedure of MOLERANKER in Algorithm 1.

## D  EXPERIMENTAL DETAILS OF MOLERANKER

In this section, we provide supplemental details on the experimental setup of MOLERANKER, covering dataset descriptions and statistics, baselines, evaluation metrics, and implementation specifics.

### D.1  DATASETS

Table 6 presents the dataset statistics, with additional details introduced as follows:

- **Sediments**: The sediment dataset comprises 98 core samples collected from four sites in Taihu Lake, the third-largest freshwater lake in China and a representative industrialized region of the Yangtze River Delta. Each core was sectioned into 1 cm intervals, freeze-dried, and analyzed using ultra-high-performance liquid chromatography coupled with high-resolution mass spectrometry (UHPLC–HRMS). From these samples, a total of 20,630 molecular features were reliably detected after quality control filtering, and 33 organophosphate esters (OPEs) were structurally annotated with high confidence. Concentrations of OPEs varied significantly across sites, with industrially influenced nearshore locations such as S3 exhibiting the highest abundance and dominance of novel OPEs. Beyond OPEs, the dataset also revealed structurally

---

**Algorithm 1** Training process of MOLERANKER

---

**Require:** MS/MS data, concentration data, PubChem database $\mathcal{D}^{pub}$, learning rate $\eta$, number of epochs $E$
**Ensure:** Optimized scoring model $f^*$
1: **for** epoch $= 1$ to $E$ **do**
2:     **Step 1: Heterogeneous Molecular Co-occurrence Graph Construction**
3:     **for** each spectrum $s_i$ **do**
4:         Estimate $m_i^{neutral}$ from precursor and fragment ions
5:         Retrieve candidate set $\mathcal{C}_i$ from $\mathcal{D}^{pub}$ via a mass tolerance filtering.
6:     **end for**
7:     Construct chemical constraints graph $\mathcal{E}_s$ using spectrum similarity
8:     Construct environmental co-occurrence graph $\mathcal{E}_c$ using concentration correlations
9:     **Step 2: Multiplex-relation Message Passing**
10:     **for** each node $v_i$ **do**
11:         Encode spectrum $e_i^s = \phi_s(s_i)$
12:         Encode candidates $\{c_{i,m}\} \rightarrow \{e_{i,m}^c\} = \phi_c(\cdot)$ and average to $\bar{e}_i^c$
13:         Form augmented node feature $z_i = \left[ \mathrm{MLP}(e_i^s) \| \mathrm{MLP}(\bar{e}_i^c) \right]$
14:     **end for**
15:     Apply relation-specific propagation over $\tilde{\mathbf{A}}_s, \tilde{\mathbf{A}}_c$ to get $z_i^{(s)}, z_i^{(c)}$
16:     Fuse with attention and residual branch to obtain $q_i$
17:     **Step 3: Dual-Tower Scoring**
18:     **for** each candidate $c_{i,m}$ **do**
19:         Compute ranking score $S_{i,m}$ based on temperature-scaled cosine similarity:
20:     **end for**
21:     Compute pairwise ranking loss $\mathcal{L}_{\mathrm{BPR}}$
22:     **Step 4: Parameter Update**
23:     Backpropagate $\nabla_\theta \mathcal{L}_{\mathrm{BPR}}$ and update model with learning rate $\eta$
24: **end for**

---

diverse, plastic-related compounds that co-occurred with OPEs in sediments. Overall, the Sediments dataset provides a detailed characterization of pollutant occurrence, spatial heterogeneity, and ecological risk in a complex freshwater environment, and serves as a valuable benchmark for developing and evaluating non-targeted screening and analog discovery methods in environmental chemistry.

- **MTBLS146** (Luan et al., 2014): The pregnancy-induced metabolic phenotype dataset consists of maternal plasma samples collected from 180 healthy women at different gestational stages. Measurements were obtained using non-targeted liquid chromatography–mass spectrometry (LC–MS) and lipidomics platforms, with quality control (QC) samples included to assess batch effects. The dataset encompasses more than one thousand metabolic features and has been widely used to investigate pregnancy-related metabolic variation as well as to benchmark normalization and batch-correction strategies in large-scale metabolomics.

- **MTBLS265** (Chaleckis et al., 2016): The human blood metabolite variability dataset comprises whole-blood samples from 30 healthy donors (15 young adults and 15 elderly) aimed at characterizing age-related differences in metabolite profiles. Data were generated using non-targeted quantitative liquid chromatography–mass spectrometry (LC–MS) on a LTQ Orbitrap Classic instrument. In total, 126 metabolites were quantified, and coefficients of variation (CV = SD/mean) were calculated across donors. Fourteen metabolites exhibited significant age-associated increases or decreases (e.g., 1,5-anhydroglucitol, dimethyl-guanosine, carnosine, $NAD^+/NADP^+$). The dataset also includes technical replicates (three injections of the same sample) and preparation replicates (three independently processed from the same blood), enabling evaluation of within-sample and preparation variability. It is frequently used in aging-metabolomics research as a benchmark for inter-individual variability in whole blood.

- **MTBLS746** (Geier et al., 2020): The spatial metabolomics dataset investigates in situ host–microbe interactions using the metaFISH pipeline, which integrates fluorescence in situ hybridization (FISH) microscopy with high-resolution mass spectrometry imaging (MSI) and ultra-performance liquid chromatography–mass spectrometry (UPLC–MS/MS). The resulting

Table 6: Statistics of the four datasets used in our experiments.

| Datasets | #Nodes | #Labels | #Train | #Val | #Test | #Avg Cand. |
|----------|--------|---------|--------|------|-------|------------|
| Sediments | 20,630 | 294 | 200 | 51 | 43 | 2,409 |
| MTBLS146 | 522 | 128 | 89 | 12 | 27 | 2,593 |
| MTBLS265 | 852 | 367 | 256 | 36 | 75 | 3,664 |
| MTBLS746 | 1,222 | 208 | 145 | 20 | 43 | 4,085 |

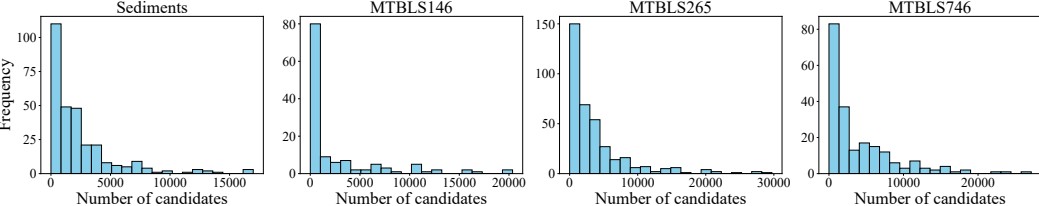

Figure 5: Candidate number distributions across datasets.

data provide spatially resolved molecular maps of small metabolites within tissue contexts, enabling the identification of compounds associated with both microbial symbionts and host cells. This dataset supports studies of chemical ecology, defensive metabolites, metabolic phenotypes, and molecular interactions at micrometer-scale resolution.

To better understand the complexity of the molecular structure ranking task, we analyze the distribution of candidate set sizes across nodes in different datasets. Figure 5 presents histograms of the number of candidates per node for Sediments, MTBLS146, MTBLS265, and MTBLS746.

## D.2 BASELINES

We use the following baselines in our experiments for a comprehensive evaluation:

- **MetFrag:** MetFrag (Ruttkies et al., 2016) is an open-source tool for annotating metabolites and other small molecules based on tandem mass spectrometry fragmentation data, representing a critical step toward structural identification. Candidate molecules are first retrieved from molecular databases using precursor mass or molecular formula, and subsequently fragmented *in silico* to generate theoretical spectra. These are then matched against experimental fragments, and the resulting fragment match scores are used to rank candidate structures. In addition, MetFrag supports extended scoring schemes, such as incorporating spectral library matches or database-specific metadata (e.g., patent and literature references), allowing flexible integration of complementary information to improve annotation accuracy.

- **CFM-ID:** CFM-ID (Wang et al., 2022) is a computational tool designed for the accurate and efficient identification of metabolites from electrospray tandem mass spectrometry (ESI-MS/MS) data. It employs Competitive Fragmentation Modeling, a probabilistic generative framework that simulates the MS/MS fragmentation process, with model parameters trained and refined using machine learning techniques. Given a target spectrum, CFM-ID predicts theoretical spectra for candidate structures and ranks them based on their similarity to the experimental spectra. For ESI-MS/MS, the input generally includes three spectra acquired at different collision energies (low/10V, medium/20V, and high/40V). Ranking is performed using similarity scores, such as Dice or DotProduct, while additional outputs include predicted chemical classes for each candidate molecule.

- **SIRIUS:** SIRIUS (Dührkop et al., 2019; 2015) is a software framework for molecular structure elucidation of metabolites and other small molecules from high-resolution LC–MS/MS data. It first combines isotope pattern analysis with fragmentation tree construction to infer candidate molecular formulas, where nodes represent fragment formulas and edges correspond to plausible fragmentation events. Candidate formulas are then ranked using isotope distributions and fragmentation consistency.

- **Random Forest (RF):** Random Forest (Rigatti, 2017) is an ensemble learning method that constructs a multitude of decision trees during training and outputs the mode of the classes for classification or mean prediction for regression of the individual trees.

- **XGBoost:** XGBoost (Chen & Guestrin, 2016) is a scalable gradient boosting framework that builds additive models in a forward stage-wise fashion and optimizes a regularized objective to control model complexity. It incorporates second-order gradient information, efficient handling of sparse data, and parallelization strategies, making it widely adopted for structured data prediction tasks.

- **MLP:** We employ a Multi-Layer Perceptron (MLP), a feedforward neural network, to model the classification task of predicting molecular structures from mass spectra. By learning non-linear mappings between spectral features and candidate molecules, MLP provides a flexible baseline for spectrum-driven molecular identification.

- **GCN:** Graph Convolutional Networks (GCN) (Kipf & Welling, 2017) extend convolutional neural networks to graph-structured data by aggregating and transforming information from neighboring nodes. Through spectral or spatial graph convolutions, GCNs capture local graph topology and node feature information, enabling effective semi-supervised learning on graph datasets.

- **GAT:** Graph Attention Networks (GAT) (Veličković et al., 2018) enhance graph representation learning by applying self-attention mechanisms to weigh the importance of neighboring nodes during message passing.

- **GraphSAGE:** GraphSAGE (Hamilton et al., 2017) is an inductive representation learning framework that generates node embeddings by sampling and aggregating features from a node's local neighborhood.

- **R-GCN:** Relational Graph Convolutional Networks (R-GCN) (Schlichtkrull et al., 2018) extend GCNs to multi-relational graphs by learning relation-specific transformation matrices and aggregating messages across typed edges.

- **HAN:** Heterogeneous Graph Attention Network (HAN) (Wang et al., 2019) exploits hierarchical attention mechanisms over both node-level neighbors and meta-path-level semantic channels. By learning meta-path-based embeddings, HAN can selectively emphasize informative relation patterns in heterogeneous graphs.

- **HetGNN:** Heterogeneous Graph Neural Network (HetGNN) (Zhang et al., 2019) models heterogeneous graphs by jointly encoding node content features and neighborhood information across different node and edge types. It employs type-specific encoders and an attentive aggregation scheme to fuse multi-type signals into unified node representations.

- **HGT:** Heterogeneous Graph Transformer (HGT) (Hu et al., 2020) generalizes the Transformer architecture to heterogeneous graphs through node- and edge-type-dependent projection matrices and relation-aware attention.

- **SeHGNN:** Simple and Efficient Heterogeneous Graph Neural Network (SeHGNN) (Yang et al., 2023) proposes a lightweight framework that decouples heterogeneous message passing into type-specific propagation and semantic aggregation.

## D.3 EVALUATION METRICS

We evaluate model performance using two widely adopted ranking metrics: Hits@K and Mean Reciprocal Rank (MRR).

**Hits@K.** Hits@K measures the proportion of cases where the ground-truth item appears in the top-$K$ predictions. Formally, let $y_i$ denote the ground-truth candidate for query $i$, and let $\hat{y}_i^{(1)}, \hat{y}_i^{(2)}, \ldots, \hat{y}_i^{(K)}$ denote the top-$K$ ranked predictions. Then:

$$\text{Hits@}K = \frac{1}{N} \sum_{i=1}^{N} \mathbb{I}\left( y_i \in \{\hat{y}_i^{(1)}, \ldots, \hat{y}_i^{(K)}\} \right), \tag{18}$$

where $\mathbb{I}(\cdot)$ is the indicator function and $N$ is the number of evaluation queries. A higher Hits@K indicates better top-$K$ retrieval accuracy. In our experiments, we set $K = 1, 10, 20$.

**Mean Reciprocal Rank (MRR).** MRR evaluates the average reciprocal rank of the ground-truth candidate in the ranked list. For each query $i$, let $\text{rank}_i$ denote the position of the ground-truth candidate in the predicted ranking. Then:

$$\text{MRR} = \frac{1}{N} \sum_{i=1}^{N} \frac{1}{\text{rank}_i}. \tag{19}$$

MRR emphasizes placing the correct candidate as high as possible in the ranking. A higher MRR reflects stronger ranking ability and early retrieval precision.

**Maximum Common Edge Subgraph (MCES@K).** Let $\tilde{\mathcal{G}}_i$ denote the ground-truth molecular graph for node $v_i$, and let $\mathcal{P}_i^{(K)}$ be the set of top-$K$ predicted molecular graphs. For each prediction $\hat{\mathcal{G}} \in \mathcal{P}_i^{(K)}$, the MCES distance $\text{MCES}(\tilde{\mathcal{G}}_i, \hat{\mathcal{G}})$ is defined as the size of the maximum common edge subgraph between $\tilde{\mathcal{G}}_i$ and $\hat{\mathcal{G}}$. We report MCES@K as:

$$\text{MCES@}K = \frac{1}{N} \sum_{i=1}^{N} \min_{\hat{\mathcal{G}} \in \mathcal{P}_i^{(K)}} \text{MCES}(\tilde{\mathcal{G}}_i, \hat{\mathcal{G}}), \tag{20}$$

where $N$ is the number of test instances. Since MCES is an edit-distance–style measure, **lower values indicate higher structural similarity**, whereas higher values reflect larger discrepancies between the generated and true molecular structures.

### D.4 IMPLEMENTATIONS

During candidate set acquisition, the mass tolerance was fixed at $\delta = 2$ parts per million (ppm). For graph construction, the thresholds $\tau_s$ and $\tau_c$ were set to $(0.1, 0.9)$ for the *Sediments*, $(0.7, 0.8)$ for *MTBLS146*, $(0.6, 0.7)$ for *MTBLS265*, and $(0.1, 0.7)$ for *MTBLS746*. Our implementation is based on PyTorch 1.8.0. All experiments are run on Nvidia RTX 4090Ti GPUs with 24GB of memory and evaluated by averaging the results over 5 runs.

## E EXPERIMENTS COMPARISON WITH DE NOVO GENERATION METHODS

We follow the MassSpecGym benchmark and include SMILES Transformer and SELFIES Transformer as two de novo generation baselines. As shown in Table 7, both models perform extremely poorly, yielding zero accuracy across all datasets. This confirms that de novo generation approaches are not appropriate when large-scale paired MS/MS–SMILES datasets are unavailable.

To further assess the quality of the generated structures, we evaluate the molecular similarity between the predictions and the ground-truth molecules using the maximum common edge subgraph (MCES) metric, which captures an edit distance over molecular graphs. Specifically, we compute the MCES distance of the closest predicted molecule within each Hits@K set to quantify how structurally similar the generated candidates are to the true compound.

As shown in Table 8, both models produce structures that diverge substantially from the ground truth. Nevertheless, the SELFIES Transformer exhibits noticeably higher similarity than the SMILES Transformer, which is consistent with established domain knowledge that SELFIES is a more robust and generation-friendly molecular representation than SMILES.

## F RUNTIME EFFICIENCY

In this section, we provide additional runtime efficiency analysis comparing machine learning baselines with MOLERANKER. As shown in Table 9, MOLERANKER achieves computational efficiency comparable to that of many machine learning baselines.

## G SUPPLEMENTAL EXPERIMENTAL RESULTS OF MOLERANKER

In this section, we provide additional experimental results of MOLERANKER, including further case studies (Figure 6) and sensitivity analyses (Figure 7, 8, 9).

Table 7: Performance comparison of two *de novo* generation methods and MoleRanker across four datasets.

| Methods | Sediments | | | MTBLS146 | | |
|---|---|---|---|---|---|---|
| | Hits@1↑ | Hits@10↑ | Hits@20↑ | Hits@1↑ | Hits@10↑ | Hits@20↑ |
| SMILES Transformer | $0.00_{\pm 0.00}$ | $0.00_{\pm 0.00}$ | $0.00_{\pm 0.00}$ | $0.00_{\pm 0.00}$ | $0.00_{\pm 0.00}$ | $0.00_{\pm 0.00}$ |
| SELFIES Transformer | $0.00_{\pm 0.00}$ | $0.00_{\pm 0.00}$ | $0.00_{\pm 0.00}$ | $0.00_{\pm 0.00}$ | $0.00_{\pm 0.00}$ | $0.00_{\pm 0.00}$ |
| MoleRanker | $\mathbf{16.74}_{\pm 0.93}$ | $\mathbf{33.02}_{\pm 2.71}$ | $\mathbf{48.84}_{\pm 2.08}$ | $\mathbf{57.04}_{\pm 1.81}$ | $\mathbf{73.33}_{\pm 2.77}$ | $\mathbf{77.04}_{\pm 1.48}$ |
| Methods | MTBLS265 | | | MTBLS746 | | |
| | Hits@1↑ | Hits@10↑ | Hits@20↑ | Hits@1↑ | Hits@10↑ | Hits@20↑ |
| SMILES Transformer | $0.00_{\pm 0.00}$ | $0.00_{\pm 0.00}$ | $0.00_{\pm 0.00}$ | $0.00_{\pm 0.00}$ | $0.00_{\pm 0.00}$ | $0.00_{\pm 0.00}$ |
| SELFIES Transformer | $0.00_{\pm 0.00}$ | $0.00_{\pm 0.00}$ | $0.00_{\pm 0.00}$ | $0.00_{\pm 0.00}$ | $0.00_{\pm 0.00}$ | $0.00_{\pm 0.00}$ |
| MoleRanker | $\mathbf{11.67}_{\pm 2.19}$ | $\mathbf{23.67}_{\pm 2.38}$ | $\mathbf{29.67}_{\pm 1.45}$ | $\mathbf{4.65}_{\pm 0.00}$ | $\mathbf{19.07}_{\pm 1.74}$ | $\mathbf{25.58}_{\pm 0.00}$ |

Table 8: MCES@K structural similarity comparison of SMILES and SELFIES Transformers across four datasets.

| Methods | Sediments | | | MTBLS146 | | |
|---|---|---|---|---|---|---|
| | MCES@1↑ | MCES@10↑ | MCES@20↑ | MCES@1↑ | MCES@10↑ | MCES@20↑ |
| SMILES Transformer | $95.85_{\pm 1.02}$ | $61.09_{\pm 1.56}$ | $40.93_{\pm 2.98}$ | $96.88_{\pm 0.50}$ | $89.00_{\pm 0.00}$ | $88.50_{\pm 0.06}$ |
| SELFIES Transformer | $28.06_{\pm 0.56}$ | $17.84_{\pm 0.06}$ | $16.66_{\pm 0.10}$ | $21.44_{\pm 0.42}$ | $12.83_{\pm 0.35}$ | $12.20_{\pm 0.10}$ |
| Methods | MTBLS265 | | | MTBLS746 | | |
| | MCES@1↑ | MCES@10↑ | MCES@20↑ | MCES@1↑ | MCES@10↑ | MCES@20↑ |
| SMILES Transformer | $98.60_{\pm 0.60}$ | $82.05_{\pm 0.53}$ | $72.35_{\pm 0.40}$ | $98.82_{\pm 0.18}$ | $89.22_{\pm 0.50}$ | $77.10_{\pm 0.40}$ |
| SELFIES Transformer | $41.70_{\pm 0.75}$ | $31.66_{\pm 0.40}$ | $30.15_{\pm 0.14}$ | $33.95_{\pm 0.24}$ | $24.65_{\pm 0.25}$ | $21.94_{\pm 0.18}$ |

Table 9: Training and inference time comparison across four datasets. We report the per-epoch training time and the inference time on the full test set (in seconds).

| Methods | Sediments | | MTBLS146 | | MTBLS265 | | MTBLS746 | |
|---|---|---|---|---|---|---|---|---|
| | training | inference | training | inference | training | inference | training | inference |
| Random Forest | – | 5.11 | – | 1.83 | – | 2.51 | – | 2.83 |
| XGBoost | – | 1.07 | – | 0.58 | – | 0.69 | – | 0.60 |
| MLP | 2.03 | 0.79 | 1.30 | 0.44 | 5.40 | 0.90 | 3.45 | 0.52 |
| GCN | 2.58 | 0.91 | 1.32 | 0.45 | 5.64 | 0.96 | 11.82 | 0.86 |
| GAT | 2.59 | 0.99 | 1.34 | 0.45 | 5.70 | 0.90 | 12.55 | 0.88 |
| GraphSAGE | 2.75 | 0.94 | 1.33 | 0.46 | 5.59 | 0.91 | 12.53 | 0.95 |
| R-GCN | 5.28 | 1.15 | 2.84 | 0.56 | 8.28 | 1.04 | 5.85 | 0.68 |
| HAN | 5.27 | 1.08 | 2.89 | 0.66 | 8.52 | 1.09 | 6.37 | 0.68 |
| HetGNN | 23.02 | 2.65 | 4.29 | 0.80 | 15.10 | 1.83 | 20.08 | 2.47 |
| HGT | 4.57 | 1.02 | 2.89 | 0.56 | 9.25 | 1.17 | 5.17 | 0.62 |
| SeHGNN | 3.94 | 0.93 | 2.51 | 0.52 | 7.33 | 1.01 | 4.39 | 0.55 |
| MoleRanker | 2.75 | 0.62 | 2.75 | 0.62 | 9.01 | 1.14 | 5.74 | 0.65 |

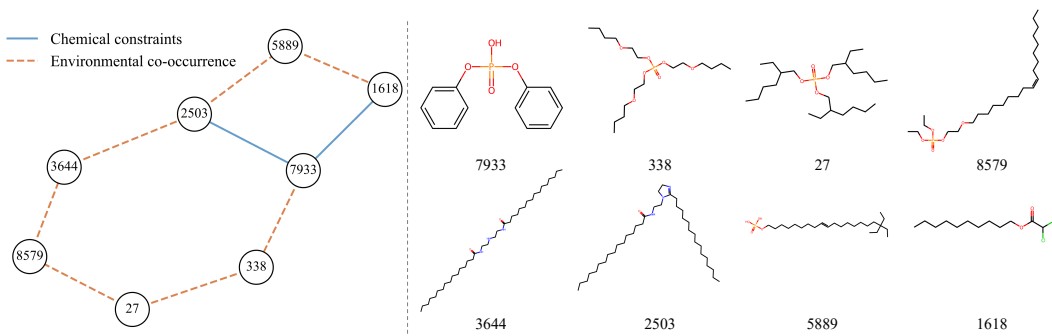

Figure 6: Additional illustration of case study. (*Left*) Supplementary heterogeneous graph from a Sediments subset. (*Right*) Supplementary MOLERANKER predictions on selected Sediments compounds.

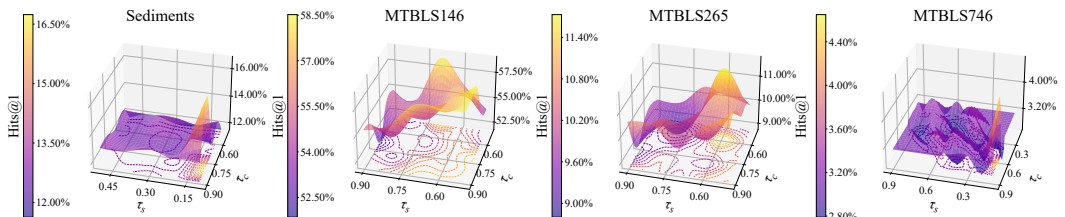

(a) Sensitivity analysis of graph construction thresholds, with results reported in Hits@1.

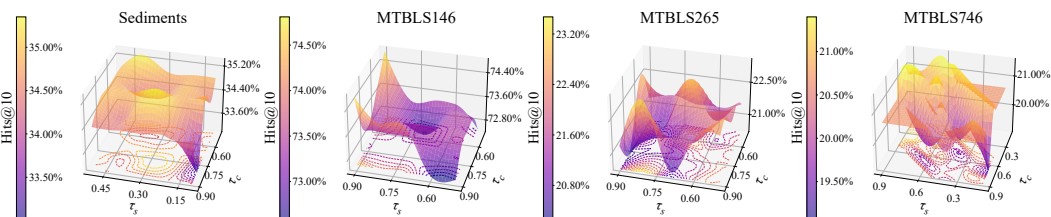

(b) Sensitivity analysis of graph construction thresholds, with results reported in Hits@10

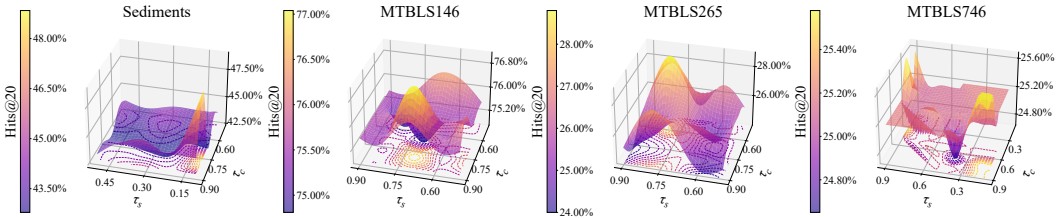

(c) Sensitivity analysis of graph construction thresholds, with results reported in Hits@20

Figure 7: Performance sensitivity of MOLERANKER under varying thresholds for chemical constraints and environmental co-occurrence. Results are reported across three evaluation metrics: (a) Hits@1, (b) Hits@10, (c) Hits@20.

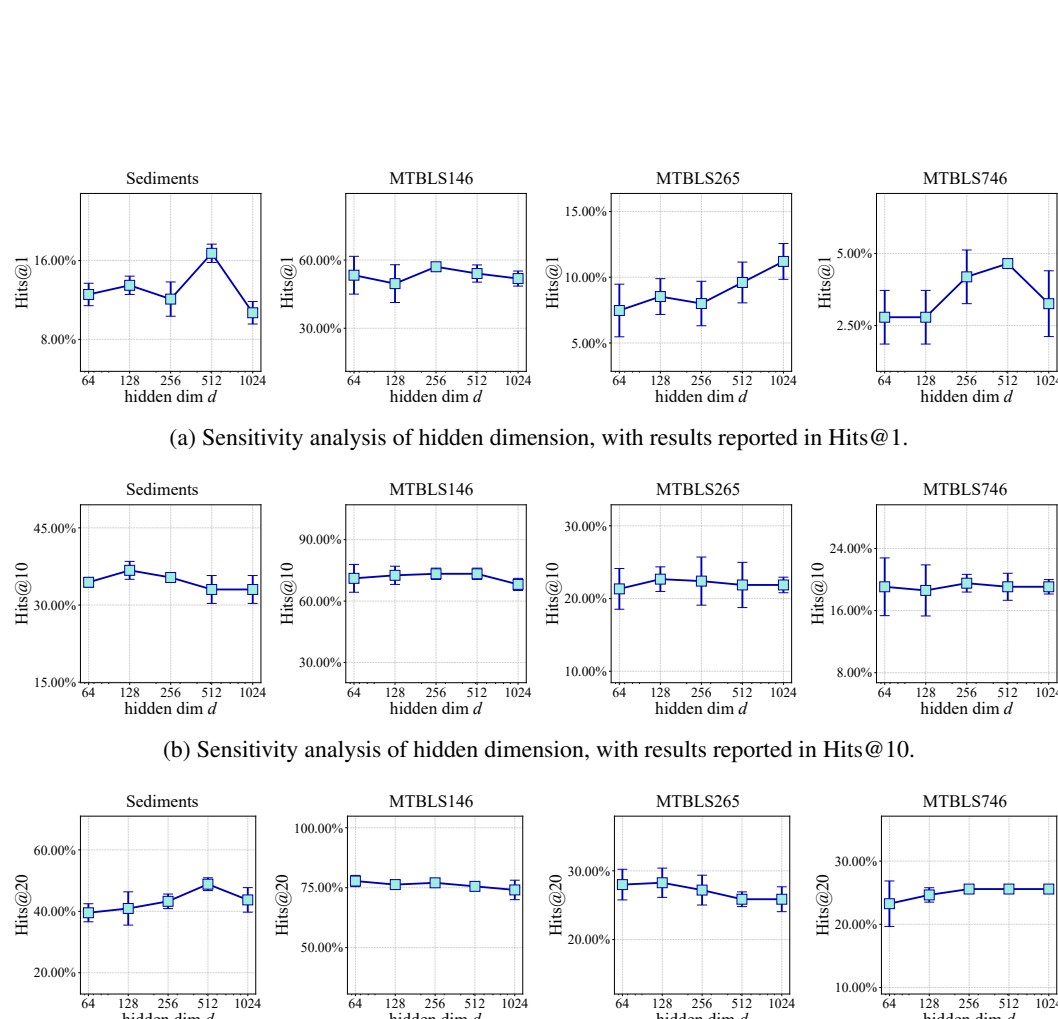

(a) Sensitivity analysis of hidden dimension, with results reported in Hits@1.

(b) Sensitivity analysis of hidden dimension, with results reported in Hits@10.

(c) Sensitivity analysis of hidden dimension, with results reported in Hits@20.

Figure 8: Hyperparameter study of MOLERANKER with respect to the size of hidden dimension $d$. Results are reported across three evaluation metrics: (a) Hits@1, (b) Hits@10, (c) Hits@20.

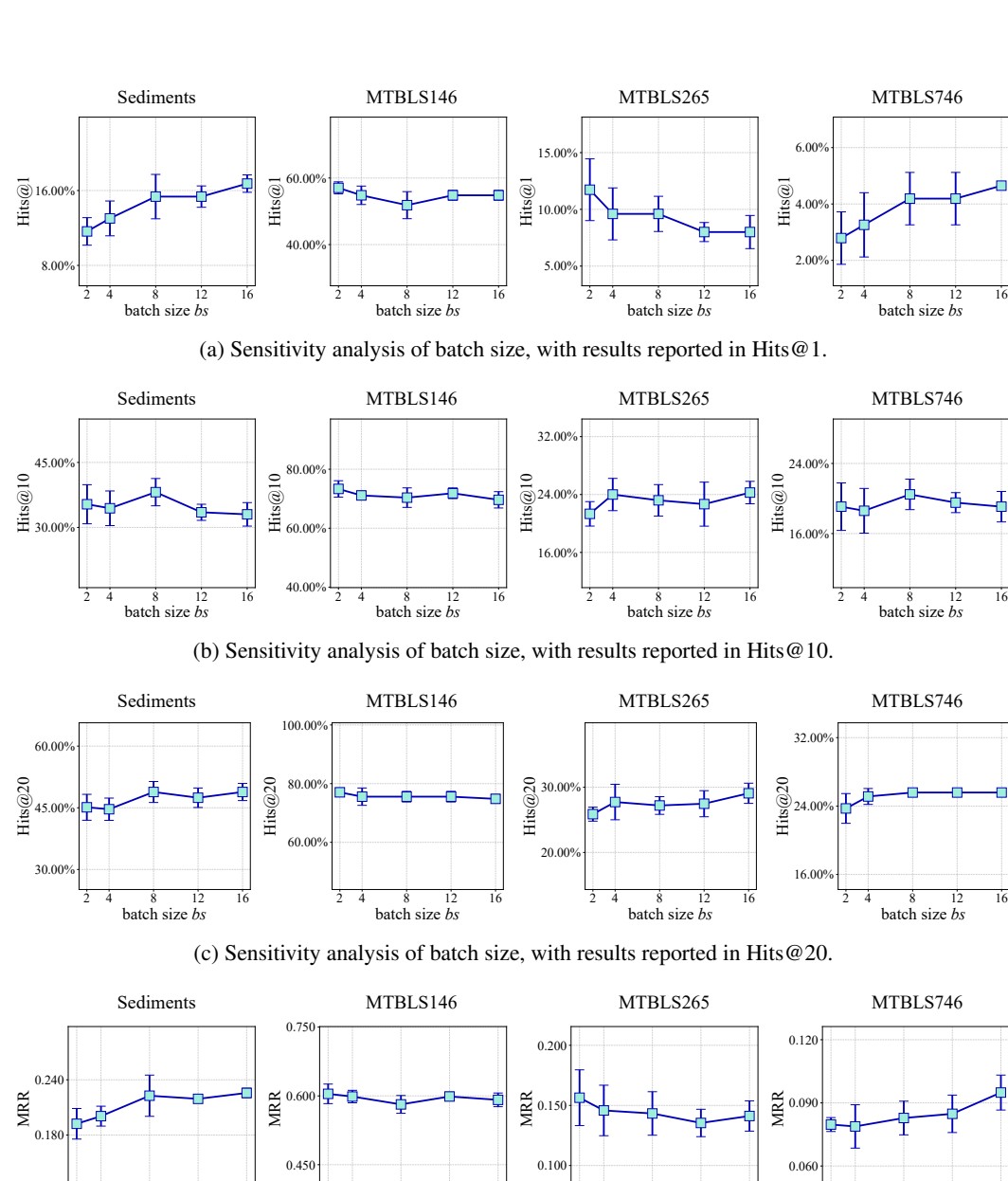

(a) Sensitivity analysis of batch size, with results reported in Hits@1.

(b) Sensitivity analysis of batch size, with results reported in Hits@10.

(c) Sensitivity analysis of batch size, with results reported in Hits@20.

(d) Sensitivity analysis of batch size, with results reported in MRR.

Figure 9: Hyperparameter study of MOLERANKER with respect to the batch size $bs$. Results are reported across three evaluation metrics: (a) Hits@1, (b) Hits@10, (c) Hits@20, (d) MRR.

