# OpenReview forum: "MoleRanker: Spectrum-Driven Molecular Structure Ranking with Heterogeneous Co-occurrence Graphs"
_ICLR.cc/2026/Conference — Submitted to ICLR 2026_

### Official Review · Reviewer_NbWy · 2025-10-30

**Soundness:** 2
**Presentation:** 2
**Contribution:** 1
**Rating:** 4
**Confidence:** 3

**Summary:**

This paper redefines molecular identification as spectrum-driven molecular structure ranking, and proposes a heterogeneous co-occurrence graph algorithm that combines molecule-level chemical concentration information and sample-level environmental information to score and rank candidate molecular structures. The authors conduct experiments on both self-collected and public datasets to demonstrate the contribution of the algorithm.

**Strengths:**

I believe the paper is clearly written and conveys the authors’ ideas well. The core idea is interesting and intuitive, the reported metrics exhibit improvements, and the ablation studies substantiate the effectiveness of both information sources.

**Weaknesses:**

1. The proposed “heterogeneous co-occurrence graph” approach does not appear to be novel; the contribution largely consists of injecting two additional information sources, while the remaining components follow standard practice. This amounts to an engineering blend rather than a deeper investigation of the molecular structure ranking problem.
2. The supplementary material lists the compared methods but omits detailed experimental configurations. I strongly recommend fully specifying all baseline setups. For instance, were the homogeneous-graph baselines (GCN, GAT, GraphSAGE) trained on the same two relation layers/graphs?
3. The training and evaluation pipeline is under-specified; please provide a detailed description.
4. Although the paper claims lower computational cost than existing methods, no supporting experiments are reported; please include computational efficiency results.

**Questions:**

1. The current comparative experiments are fairly conventional. Was a comprehensive investigation of strong baselines conducted?
2. The reported performance of MetFrag, CFM-ID, and SIRIUS is near zero on three datasets. Is this consistent with prior empirical evidence? Please provide a detailed explanation for these results.

---

> ### Author Response · Authors · 2025-11-20
> **Response to Reviewer NbWy**
>
> Thank you for your valuable suggestions. For clarity, we provide point-by-point responses to your comments as follows:
> - **W1**
>
> The core contribution of our work is **an AI for Science effort that reformulates real-world molecular identification as a tractable molecular structure ranking task.** To support this problem, **we curate new datasets and develop MoleRanker as a domain-informed framework grounded in established chemical principles.**
>
> The heterogeneous co-occurrence graph and multiplex-relation message passing are **motivated by two well-supported scientific observations**:
> (i) compounds with similar molecular structures tend to exhibit higher spectral similarity (*chemical constraints*), and
> (ii) compounds sharing functional groups often show correlated concentration levels in environmental samples (*environmental co-occurrence*).
>
> These domain principles directly guide the model design and enable the extraction of meaningful molecular relations that traditional approaches cannot capture. Therefore, **MoleRanker is not an engineering blend.**
> - **W2**
>
> We clarify that the homogeneous-graph baselines (GCN, GAT, GraphSAGE) **were not trained on the two relation graphs used in MoleRanker**. Instead, **they were trained solely on the standard GNPS-style molecular network**, which contains a single relation based on spectral similarity.
> - **W3**
>
> The dataset splits, baseline methods, and evaluation metrics are specified in Section 5.1 and Appendix D. **Here we clarify the training and evaluation pipeline.**
>
> *For MetFrag, CFM-ID, and SIRIUS*, we use the publicly released tools and run them on the test sets with their standard configurations. *For Random Forest, XGBoost, and MLP*, we form training pairs by combining the spectral representation with the structural embedding of each candidate molecule, and train these models as binary classifiers over candidate sets. *For GCN, GAT, and GraphSAGE*, we fuse the spectral representation with the aggregated structural representation of the candidates, perform message passing on the traditional GNPS-style molecular network built from spectral similarity, and obtain candidate scores using the same dual-tower scoring mechanism as MoleRanker.
>
> An anonymous code link is also provided in the manuscript, where the complete training and evaluation pipelines can be examined in detail.
> - **W4**
>
> We include **a comprehensive comparison of computational efficiency across all methods**. Specifically, we report the inference runtime of MoleRanker against the traditional molecular identification tools (MetFrag, CFM-ID, and SIRIUS) on all four datasets, and MoleRanker is substantially faster in every case. We also provide per-iteration training time and inference time for MoleRanker alongside the machine-learning baselines, showing that their computational costs are comparable. These results demonstrate that **MoleRanker achieves strong performance without introducing additional computational overhead.**
>
> |Method|Sediments|MTBLS146|MTBLS265|MTBLS746|
> |-|-|-|-|-|
> |MetFrag|2.5h|3min|1.2h|16min|
> |CFM-ID|36h|109h|89h|99h|
> |SIRIUS|2min|3min|3h14min|3min|
> |MoleRanker|**0.62s**|**0.62s**|**1.14s**|**0.65s**|
>
> |Methods|Sediments Training|Sediments Inference|MTBLS146 Training|MTBLS146 Inference|MTBLS265 Training|MTBLS265 Inference|MTBLS746 Training|MTBLS746 Inference|
> |-|-|-|-|-|-|-|-|-|
> |RandomForest|-|5.11|-|1.83|-|2.51|-|2.83|
> |XGBoost|-|1.07|-|0.58|-|0.69|-|0.60|
> |MLP|2.03|0.79|1.30|0.44|5.40|0.90|3.45|0.52|
> |GCN|2.58|0.91|1.32|0.45|5.64|0.96|11.82|0.86|
> |GAT|2.59|0.99|1.34|0.45|5.70|0.90|12.55|0.88|
> |GraphSAGE|2.75|0.94|1.33|0.46|5.59|0.91|12.53|0.95|
> |MoleRanker|2.75|0.62|2.75|0.62|9.01|1.14|5.74|0.65|
> - **Q1**
>
> Yes, we conducted a comprehensive investigation of potential baselines. Since our work introduces a new task formulation that reframes molecular identification as a molecular structure ranking problem, **the most relevant baselines are the tools actually used in practice for MS/MS-based identification**, namely MetFrag, CFM-ID, and SIRIUS. We also include classical machine-learning models to isolate the contribution of our heterogeneous molecular co-occurrence graph. While recent deep learning approaches exist, they rely on large quantities of high-quality paired MS/MS–SMILES data, **which are not available in real-world settings due to the high cost of data acquisition and expert curation**. Consequently, such methods cannot be reliably trained or applied in our experiments.
> - **Q2**
>
> We appreciate the reviewer’s concern regarding the very low performance of MetFrag, CFM-ID, and SIRIUS on three of our datasets, and we confirm that these results accurately reflect our experimental observations. A key reason for the near-zero performance is that **these tools frequently assign identical scores to many distinct candidate structures**, which would otherwise inflate Hits@K. **In our evaluation protocol, such cases are treated as failures.**

---

> > ### Author Response · Authors · 2025-12-03
> >
> > Due to an unexpected accident, we were unable to engage in discussion with you during the rebuttal stage. Nevertheless, we have carefully considered your valuable suggestions and incorporated detailed revisions into the updated manuscript. We hope that these improvements help clarify our contributions and provide a clearer understanding of our work. Thank you.

---

### Official Review · Reviewer_8Hq5 · 2025-10-30

**Soundness:** 2
**Presentation:** 2
**Contribution:** 2
**Rating:** 4
**Confidence:** 4

**Summary:**

This paper presents MOLERANKER, a heterogeneous graph neural network specifically designed for spectrum-based molecular structure ranking. The core idea is to integrate chemical constraints (spectral similarity) and environmental co-occurrence (concentration correlations between samples) into multiple heterogeneous graphs, thereby enhancing message passing and improving the accuracy of candidate molecule ranking. The proposed architecture comprises spectrum-based and SMILES-based encoders, relation-aware graph convolution, and a dual-tower scoring mechanism optimized via a pairwise Bayesian Personalized Ranking (BPR) loss function. Experimental results on four datasets—including a newly curated environmental pollutant dataset and three human metabolomics datasets—demonstrate that the proposed method significantly outperforms existing approaches.

**Strengths:**

1. The integration of a dual-tower scoring mechanism with a pairwise Bayesian Personalized Ranking (BPR) objective exhibits practical value for large candidate sets and severe class imbalance problems.
2. The experiments conducted in this study are relatively comprehensive, and the results are favorable.
3. The visualization outcomes are presented with commendable clarity.

**Weaknesses:**

1. Omission of certain recent baselines in evaluation: The selection of baselines overlooks several directly relevant recent methods, particularly those leveraging topological or spectral graph modeling, or explicitly employing multi-relational molecular graphs for ranking tasks beyond the GNPS ecosystem.
2. Insufficient discussion of practical limitations and generalization ability: Although the paper demonstrates strong performance on the relevant datasets, the discussion does not fully address potential generalization constraints—such as distributional shift in environmental chemistry or biomedical metabolomics—nor does it quantify the effects of batch variability, sample bias, or differences in candidate quality.
3. Lack of, or insufficient, justification for not adopting other established GNN variants or loss functions: While the study implements standard GCN, GAT, and GraphSAGE as baselines, it does not utilize other advanced heterogeneous/relational GNN variants that have proven effective for multi-graph or biochemical graph tasks. The rationale for including only the proposed baselines is not sufficiently substantiated.
4. Limited discussion on computational efficiency and scalability: Although the manuscript claims that MOLERANKER surpasses spectral isolation methods in computational efficiency, it provides no runtime or complexity analysis. Given the scale of candidate datasets shown in Figure 5 and Table 4, a clearer comparison is needed to articulate the trade-offs between actual runtime and scalability.

**Questions:**

See the weaknesses.

---

> ### Author Response · Authors · 2025-11-20
> **Response to Reviewer 8Hq5**
>
> Thanks for your constructive suggestions. For clarity, we provide point-by-point responses to your questions as follows:
> - **Response to W1**
>
> **We conducted an extensive survey of recent molecular identification methods and found that they are not suitable as baselines for our setting.** These approaches typically require large-scale, high-quality paired MS/MS–SMILES datasets to train deep generative or contrastive models, but such data are not available in real-world identification workflows due to the high cost of acquisition and expert curation. *Therefore, they cannot be reliably trained or deployed in our scenario.*
>
> In our experiments, we compare against MetFrag, CFM-ID, and SIRIUS, which are the most widely used and practically deployable baselines in biochemical and environmental analysis. To further assess the benefit of incorporating chemical constraints and environmental co-occurrence, we also include classical GNN models trained on traditional GNPS-style molecular networks.
>
> - **Response to W2**
>
> We agree that distributional shift, batch variability, sample bias, and candidate-set quality are important considerations for real-world molecular identification. **However, these issues do not apply to our experimental setting for the following reasons.**
>
> *(1) No batch variability or sample bias.*
> All MS/MS spectra in our study were acquired **under uniform laboratory conditions using identical instrumentation and collision settings.** Therefore, the datasets do not exhibit cross-batch variation or heterogeneous sample sources that could introduce sample bias.
>
> *(2) Unbiased candidate generation.*
> **Candidate molecules are retrieved strictly through mass-tolerance filtering on PubChem, the most comprehensive chemical database.** This avoids manual curation or class-specific heuristics and ensures consistent candidate quality across samples.
>
> *(3) Generalization validated within realistic constraints.*
> MoleRanker is evaluated on four datasets from different experimental contexts, **showing consistent effectiveness across varying molecular distributions.** Cross-domain generalization is indeed a more challenging setting, but it requires large quantities of labeled MS/MS–structure pairs that are not available in practice. Conducting such experiments without adequate labeled data would not reflect realistic workflows and may lead to misleading conclusions.
>
> - **Response to W3**
>
> We clarify that the primary objective of our work is not to introduce a new heterogeneous GNN variant, but to address a real scientific problem by reformulating molecular identification as a molecular structure ranking task. Therefore, **the most relevant baselines are the tools actually used in practice for MS/MS-based identification**, namely MetFrag, CFM-ID, and SIRIUS.
>
> We additionally include standard GNN models (GCN, GAT, GraphSAGE) to isolate the benefit of incorporating chemical constraints and environmental co-occurrence beyond traditional molecular networks. **While many advanced heterogeneous or relational GNN variants exist, they are not directly applicable to the domain-specific workflow of MS/MS-based candidate ranking and rely on assumptions that do not hold in this setting.**
>
> For these reasons, we believe the chosen baselines are the most appropriate and scientifically justified for evaluating MoleRanker.
> - **Response to W4**
>
> We provide a detailed comparison of the inference runtime across MoleRanker and traditional molecular identification tools (MetFrag, CFM-ID, and SIRIUS) on all four datasets. As shown in the table, **MoleRanker achieves substantially faster inference than all three traditional methods.** We further report the per-iteration training time (s/epoch) and inference time (s) for MoleRanker and other machine learning baselines. The results show that **their computational costs are comparable**, indicating that MoleRanker does not introduce additional overhead relative to standard ML models. **Overall, MoleRanker remains the most effective and practical approach among all baselines.**
>
> |Method|Sediments|MTBLS146|MTBLS265|MTBLS746|
> |-|-|-|-|-|
> |MetFrag|2.5h|3min|1.2h|16min|
> |CFM-ID|36h|109h|89h|99h|
> |SIRIUS|2min|3min|3h14min|3min|
> |MoleRanker|**0.62s**|**0.62s**|**1.14s**|**0.65s**|
>
> |Methods|Sediments Training|Sediments Inference|MTBLS146 Training|MTBLS146 Inference|MTBLS265 Training|MTBLS265 Inference|MTBLS746 Training|MTBLS746 Inference|
> |-|-|-|-|-|-|-|-|-|
> |RandomForest|-|5.11|-|1.83|-|2.51|-|2.83|
> |XGBoost|-|1.07|-|0.58|-|0.69|-|0.60|
> |MLP|2.03|0.79|1.30|0.44|5.40|0.90|3.45|0.52|
> |GCN|2.58|0.91|1.32|0.45|5.64|0.96|11.82|0.86|
> |GAT|2.59|0.99|1.34|0.45|5.70|0.90|12.55|0.88|
> |GraphSAGE|2.75|0.94|1.33|0.46|5.59|0.91|12.53|0.95|
> |MoleRanker|2.75|0.62|2.75|0.62|9.01|1.14|5.74|0.65|

---

> > ### Author Response · Authors · 2025-12-02
> > **Supplemental Response to W3**
> >
> > In light of the thoughtful concerns about baseline selection, we include **5 additional variants of graph neural networks** (R-GCN[1], HAN[2], HetGNN[3], HGT[4], and SeHGNN[5]) by adapting them to our spectrum-driven molecular structure ranking task.
> >
> > The experimental results show that MoleRanker continues to exhibit superior performance compared with other heterogeneous GNN models, indicating that our model architecture is better suited for capturing chemical constraints and environmental co-occurrence among molecules.
> >
> > Although an unexpected accident prevented us from engaging in discussion with you during the rebuttal stage, we hope that this response provides additional clarity regarding our work and supports your understanding. Thank you.
> >
> > |Methods| Sediments Hits@1  | Sediments Hits@10 | Sediments Hits@20| Sediments MRR| MTBLS146 Hits@1         | MTBLS146 Hits@10| MTBLS146 Hits@20 | MTBLS146 MRR               |
> > | - | - | - | - | - | - | -| - | - |
> > | R-GCN      | 11.63 ± 1.47           | 35.81 ± 2.37     | 45.12 ± 1.86           | 0.1948 ± 0.0101           | 53.33 ± 5.54           | 70.37 ± 4.06           | 73.33 ± 4.32           | 0.5769 ± 0.0497           |
> > | HAN        | 11.16 ± 0.93           | 33.02 ± 1.43     | 43.26 ± 2.37           | 0.1861 ± 0.0086           | 52.59 ± 4.32           | 68.15 ± 3.78           | 72.59 ± 1.81           | 0.5681 ± 0.0343           |
> > | HetGNN     | 12.56 ± 2.37           | 36.28 ± 2.37     | 44.19 ± 2.94           | 0.1989 ± 0.0195           | 48.89 ± 3.63           | 68.89 ± 3.78           | 72.59 ± 5.02           | 0.5578 ± 0.0251           |
> > | HGT        | 12.56 ± 2.37           | 34.88 ± 0.00     | 43.26 ± 2.37           | 0.1970 ± 0.0160           | 50.37 ± 3.78           | 72.59 ± 1.81           | 74.07 ± 2.34           | 0.5627 ± 0.0258           |
> > | SeHGNN     | 11.16 ± 0.93           | 37.21 ± 4.88     | 44.65 ± 2.71           | 0.1930 ± 0.0107           | 51.85 ± 4.06           | 70.37 ± 3.31           | 74.07 ± 2.34           | 0.5700 ± 0.0316           |
> > | MoleRanker | **16.74 ± 0.93** | 33.02 ± 2.71     | **48.84 ± 2.08** | **0.2257 ± 0.0052** | **57.04 ± 1.81** | **73.33 ± 2.77** | **77.04 ± 1.48** | **0.6044 ± 0.0214** |
> >
> > | Methods    | MTBLS265 Hits@1         | MTBLS265 Hits@10        | MTBLS265 Hits@20        | MTBLS265 MRR               | MTBLS746 Hits@1        | MTBLS746 Hits@10 | MTBLS746 Hits@20        | MTBLS746 MRR |
> > | - | - | - | - | - | - | - | - | - |
> > | R-GCN      | 8.80 ± 1.60            | 22.67 ± 1.46           | 29.07 ± 2.13           | 0.1416 ± 0.0134           | 3.26 ± 1.14           | 19.53 ± 1.14    | 25.58 ± 0.00           | 0.0838 ± 0.0030|
> > | HAN        | 7.73 ± 1.00 | 22.67 ± 1.89| 29.07 ± 2.13           | 0.1300 ± 0.0064 | 4.19 ± 0.93           | 19.53 ± 1.14    | 25.12 ± 0.93 | 0.0903 ± 0.0068 |
> > | HetGNN| 6.13 ± 0.65| 21.07 ± 2.29 | 26.40 ± 3.71| 0.1135 ± 0.0055| 4.65 ± 0.00           | 19.07 ± 1.74    | 25.58 ± 0.00           | 0.0936 ± 0.0090  |
> > | HGT | 8.00 ± 0.84 | 22.67 ± 1.46 | 26.93 ± 2.44  | 0.1298 ± 0.0092 | 4.19 ± 0.93| 20.47 ± 0.93    | 25.58 ± 0.00 | 0.0906 ± 0.0053  |
> > | SeHGNN     | 9.60 ± 1.00| 21.87 ± 2.75 | 27.47 ± 2.17  | 0.1398 ± 0.0145 | 3.26 ± 1.14  | 20.00 ± 2.37| 25.58 ± 0.00 | 0.0882 ± 0.0064|
> > | MoleRanker | **11.67 ± 2.19** | **23.67 ± 2.38** | **29.67 ± 1.45** | **0.1567 ± 0.0130** | **4.65 ± 0.00** | 19.07 ± 1.74|**25.58 ± 0.00**|**0.0949 ± 0.0083**|
> >
> > [1] Modeling Relational Data with Graph Convolutional Networks, European semantic web conference, 2018.
> >
> > [2] Heterogeneous Graph Attention Network, WWW, 2019.
> >
> > [3] Heterogeneous Graph Neural Network, KDD, 2019.
> >
> > [4] Heterogeneous Graph Transformer, WWW, 2020.
> >
> > [5] Simple and Efficient Heterogeneous Graph Neural Network, AAAI, 2023.

---

### Official Review · Reviewer_sthq · 2025-10-31

**Soundness:** 3
**Presentation:** 3
**Contribution:** 2
**Rating:** 2
**Confidence:** 2

**Summary:**

The paper proposes a new method of identifying molecules from mass spectra combined with their co-occurrences. The work combines information from multiple resources, including correlation information computed from multiple samples, encoding of mass spectra, similarity between mass spectra, and molecule structures of candidates. The proposed method designs a graph neural network to learn information from all these resources. By minimizing the loss derived from a few mass spectra with known molecules, the model learns to identify molecule structures of other mass spectra. The performance shows that the proposed model achieves good performance in the transductive setting (identifying molecules in the constructed graph on all mass spectra).

**Strengths:**

On the data with multiple samples, it is a novel approach to include the correlation information from the concentration of molecules to improve the model's performance.

The proposed method outperforms several baselines, though new baselines from recent years should be included.

**Weaknesses:**

1. The setup of the problem is not very realistic. The research seems to be a rediscovery of the ground truth. In a real situation, we can obtain a collection of mass spectra and construct a graph over them. However, it is hard to know the molecular structures of some nodes for training a neural network. It is possible that we could get the structures of a few of them because they are easy to identify, or we can use some external information. Even in this case, the known molecules will not represent the distribution of all molecules in the graph. The training model might be biased. Without the ability to generalize to different networks, it is hard to put the model to use in real situations.

2. The model design with graph neural networks is largely known to the community. Therefore, the contribution to the machine learning aspect is limited.

3. In the experiment, the three algorithms specially designed for molecule identification, MetFrag, CMF-ID, SIRIUS, are traditional methods without advanced neural networks. Other baselines are all generic learning models. There are a series of improvements [1] over these three methods, and these new methods should be included in the comparison.


[1] Liu, Youzhong, et al. "Current and future deep learning algorithms for tandem mass spectrometry (MS/MS)‐based small molecule structure elucidation." Rapid Communications in Mass Spectrometry 39 (2025): e9120.

**Questions:**

I have no questions.

---

> ### Author Response · Authors · 2025-11-20
> **Response to Reviewer sthq**
>
> Thank you for your valuable suggestions. For clarity, we provide point-by-point responses to your questions as follows:
>
> - **Response to W1**
>
> We appreciate the reviewer’s concern about the realism of the problem setup. In real-world settings, **it is extremely difficult for scientists to obtain large-scale, high-quality pairs of mass spectra and molecular structures under the same experimental environment**. Consequently, constructing a massive fully-labeled dataset for training de novo generation models is currently unrealistic. In fact, prior work has shown that *even when using the MassSpecGym[1] benchmark (with 231k spectra and 29k molecular structures), de novo generation models still achieve essentially **0% Hits@1 accuracy**, highlighting the intrinsic difficulty of the task.*
>
> Given the fundamental challenges of molecular identification, and considering that in practice scientists usually narrow the search space to plausible candidates (using prior knowledge of chemical classes, biosynthetic pathways, and sample context) before ranking the most likely structure, **we choose to formulate our problem as a molecular structure ranking task**. We believe that this formulation more faithfully reflects how molecular identification is actually performed in real laboratories.
>
> [1] MassSpecGym: A benchmark for the discovery and identification of molecules, NeurIPS 2024.
>
> - **Response to W2**
>
> The primary contribution of our work is not merely to propose a new GNN architecture, but rather to **bridge a real scientific problem in molecular identification with a feasible machine-learning problem**. Specifically, we model molecular structure identification as a molecular structure ranking task, which reflects how scientists approach the problem in real-world settings. **We also curate four new datasets to support this task**, enabling ML researchers to study and benchmark models under realistic constraints. Furthermore, **our framework is inspired by two empirically grounded phenomena**:
> (i) chemically similar compounds tend to exhibit higher spectral similarity (which we refer to as *chemical constraints*), and
> (ii) compounds sharing similar functional groups often show correlated concentration patterns in real samples (referred to as *environmental co-occurrence*).
> These observations motivate us to construct a heterogeneous molecular co-occurrence graph and design MoleRanker to effectively exploit these relationships to enhance structure prediction for unknown compounds.
>
> We believe **this AI-for-Science effort will encourage more ML researchers to engage with real-world molecular identification problems and inspire more advanced models for this domain**, while also demonstrating to the chemistry and environmental science communities the value of ML approaches for addressing core scientific challenges.
>
> - **Response to W3**
>
> As summarized in the conclusion of the referenced paper:
> > given enough training data, adapted DL architectures, optimal hyperparameters and computing power, DL frameworks can predict small molecule structures, completely or at least partially, from MS/MS spectra.
>
> However, **these assumptions do not hold in the real-world molecular identification scenario we study**. In practice, obtaining large-scale high-quality paired MS/MS–SMILES data is extremely costly due to the need for controlled laboratory acquisition, instrument calibration, expert curation, and chemical standard availability. Consequently, the large-scale annotated datasets required to train recent deep learning generative models are not available in our setting, and these methods cannot be reliably trained or reproduced.
>
> For these reasons, although the emerging DL models are of great interest, **they cannot be included as comparable baselines within our experimental framework**. Therefore, we use MetFrag, CFM-ID, and SIRIUS as the strongest available traditional mass spectrometry baselines for our setting.

---

> > ### Comment · Reviewer_sthq · 2025-11-20
> > **the concern about training the model with labeled nodes on a graph**
> >
> > Thank you for your response. I think my main consider is not about using the true labels (molecule structures) of mass spectra. Instead, the concern is about the random selection of graph nodes as the training set. Then the assumption is that a random set of graph nodes has labels. This is unlikely to be true in reality, which means that the label propagation through the graph neural network won't work.
> >
> > Your response is mostly about the case without a graph. This is not a problem because the neural network can generalize from mass spectra from one application to those from another application. However, once you impose the graph on one application, the (graph) neural network will lose its generalizability.

---

> > > ### Author Response · Authors · 2025-11-20
> > > **Response to the concern about training the model with labeled nodes on a graph**
> > >
> > > Thank you for the further clarification of your concerns.
> > >
> > > Firstly, we would like to clarify the setup regarding training data selection and our model objective:
> > >
> > > - The random split into train/validation/test sets is **performed exclusively on nodes that possess ground-truth labels** (i.e., mass spectra whose true molecular structure is known and included in the candidate set). This is a standard and necessary procedure to **establish a reproducible and evaluable training–testing pipeline**.
> > > - **MoleRanker is a learning-to-rank model rather than a label-propagation model.** Specifically, every node in the graph (both labeled and unlabeled in the real world) has a mass spectrum and an associated set of candidate molecular structures. The labeled nodes provide the supervision needed to learn the ranking of the correct structure within each candidate set, and we leverage node–node relationships in the heterogeneous graph to enhance this ranking function.
> > >
> > > Moreover, our setup closely reflects real laboratory practice. Only a subset of spectra can be confidently identified, while most remain unknown. Scientists routinely use relationships among spectra to reason about unidentified compounds. **MoleRanker is explicitly designed to mimic this workflow by combining supervised ranking with graph-based contextual information.**
> > >
> > > Secondly, regarding the concern that incorporating graph structure may hinder cross-domain generalization, **we clarify that such generalization is currently not feasible for molecular identification.** Even within a single domain, existing approaches, including those trained on large public datasets such as MassSpecGym, still yield 0% Hits@1 accuracy for molecular identification. **Therefore, cross-domain generalization is an even more difficult setting.** reformulating the problem as molecular structure ranking is reasonable and well-justified. The introduction of the heterogeneous graph is **inspired by two well-supported scientific observations** (chemical constraints and environmental co-occurrence), and our experimental results demonstrate that **leveraging the heterogeneous molecular co-occurrence graph substantially improves ranking accuracy within each dataset.**

---

> ### Author Response · Authors · 2025-12-02
> **Supplemental Response to Reviewer sthq**
>
> Due to an unexpected accident, we are unable to engage in discussion with you during the rebuttal stage. We hope that our previous response has already addressed your concerns. The supplemental response provided here aims to more directly explain our rationale for baseline selection. Our initial decision not to incorporate certain methods is based on practical considerations: *de novo* generation models require large amounts of high-quality paired MS/MS–SMILES data for training, and datasets of such scale are rarely feasible in real-world scientific scenarios.
>
> To more directly address your concern, we follow the MassSpecGym benchmark[1] and include **SMILES Transformer** and **SELFIES Transformer** as two advanced baselines. As expected, the results confirm that ***de novo* generation models are not suitable when large-scale paired MS/MS–SMILES data are unavailable.** This finding is fully consistent with the results reported in MassSpecGym, where even with 231K spectra and 29K SMILES molecules, the *de novo* models still achieve a Hits@1 score of 0. In addition, we incorporate several heterogeneous graph neural network variants into our spectrum-driven molecular structure ranking task. These results are provided in our response to Reviewer `8Hq5`.
>
> Overall, we would like to clarify that our baseline selection strategy is grounded in extensive literature investigation and discussions with domain experts. Based on these efforts, we identify a total of **17 baselines**, including **3 widely used science-domain baselines and 14 machine learning methods**, and we conduct reproducible and fair experimental evaluations across all of them.
>
> | Methods             | Sediments Hits@1 | Sediments Hits@10 | Sediments Hits@20 | MTBLS146 Hits@1 | MTBLS146 Hits@10 | MTBLS146 Hits@20 |
> |--------------------|------------------|-------------------|-------------------|------------------|------------------|------------------|
> | SMILES Transformer  | 0.00 ± 0.00      | 0.00 ± 0.00       | 0.00 ± 0.00       | 0.00 ± 0.00      | 0.00 ± 0.00      | 0.00 ± 0.00      |
> | SELFIES Transformer | 0.00 ± 0.00      | 0.00 ± 0.00       | 0.00 ± 0.00       | 0.00 ± 0.00      | 0.00 ± 0.00      | 0.00 ± 0.00      |
> | MoleRanker          | **16.74 ± 0.93** | **33.02 ± 2.71**  | **48.84 ± 2.08**  | **57.04 ± 1.81** | **73.33 ± 2.77** | **77.04 ± 1.48** |
>
> |Methods|MTBLS265 Hits@1| MTBLS265 Hits@10|MTBLS265 Hits@20|MTBLS746 Hits@1|MTBLS746 Hits@10|MTBLS746 Hits@20|
> | - | - | - | -| - | - | - |
> | SMILES Transformer | 0.00 ± 0.00 | 0.00 ± 0.00     |0.00 ± 0.00 | 0.00 ± 0.00 |0.00 ± 0.00 | 0.00 ± 0.00 |
> | SELFIES Transformer | 0.00 ± 0.00 | 0.00 ± 0.00    |0.00 ± 0.00| 0.00 ± 0.00 | 0.00 ± 0.00| 0.00 ± 0.00 |
> | MoleRanker | **11.67 ± 2.19** | **23.67 ± 2.38** | **29.67 ± 1.45** | **4.65 ± 0.00** | **19.07 ± 1.74**|**25.58 ± 0.00**|
>
> We further evaluate the structural similarity between the generated molecules and the ground-truth structures using the **maximum common edge subgraph (MCES)** metric, which measures an edit distance over molecular graphs. Specifically, we assess how close the top-K predicted molecules are to the true structure by computing the MCES distance for the most similar prediction within each Hits@K set.
>
> The results show that both models generate molecules that deviate substantially from the true structures. However, SELFIES Transformer achieves noticeably better similarity than SMILES Transformer. This observation aligns with the domain consensus that SELFIES representations are more robust and better suited for molecular generation than SMILES. **These findings further demonstrate the rigor and reliability of our experiments.**
>
> | Methods            | Sediments MCES@1            | Sediments MCES@10           | Sediments MCES@20           | MTBLS146 MCES@1           | MTBLS146 MCES@10         | MTBLS146 MCES@20         |
> |--------------------|----------------|----------------|----------------|---------------|--------------|--------------|
> | SMILES Transformer | 95.85 ± 1.02   | 61.09 ± 1.56   | 40.93 ± 2.98   | 96.88 ± 0.50  | 89.00 ± 0.00 | 88.50 ± 0.06 |
> | SELFIES Transformer| 28.06 ± 0.56   | 17.84 ± 0.06   | 16.66 ± 0.10   | 21.44 ± 0.42  | 12.83 ± 0.35 | 12.20 ± 0.10 |
>
> | Methods            | MTBLS265 MCES@1           | MTBLS265 MCES@10          | MTBLS265 MCES@20          | MTBLS746 MCES@1           | MTBLS746 MCES@10         | MTBLS746 MCES@20         |
> |--------------------|---------------|--------------|--------------|---------------|--------------|--------------|
> | SMILES Transformer | 98.60 ± 0.60  | 82.05 ± 0.53 | 72.35 ± 0.40 | 98.82 ± 0.18  | 89.22 ± 0.50 | 77.10 ± 0.40 |
> | SELFIES Transformer| 41.70 ± 0.75  | 31.66 ± 0.40 | 30.15 ± 0.14 | 33.95 ± 0.24  | 24.65 ± 0.25 | 21.94 ± 0.18 |
>
> [1] MassSpecGym: A benchmark for the discovery and identification of molecules, NeurIPS, 2024.

---

### Official Review · Reviewer_9kL1 · 2025-10-31

**Soundness:** 3
**Presentation:** 3
**Contribution:** 3
**Rating:** 6
**Confidence:** 3

**Summary:**

The authors address the problem molecular structure ranking from tandem mass spectrometry data. They present MoleRanker, a novel graph neural network-based method which uses molecular groupings (as graphs) of shared functional groups (environmental co-occurrence) and chemical constraints. The authors additionally construct a new dataset to evaluate their proposed approach. Through a set of empirical experiments across several datasets, comparing with various baselines, and numerous ablations, the authors demonstrate the effectiveness of their proposed method for spectrum-driven molecular structure ranking.

**Strengths:**

- The paper is clearly written and well presented (barring some minor comments below). As a result, the claims and contributions of this work are easy to follow.
- The authors construct a new tandem mass spectrometry dataset to evaluate and benchmark molecular identification as a spectrum-driven molecular structure ranking task.
- The authors introduce a novel method for molecular identification and validate it through comprehensive empirical experiments. The newly proposed method offers a novel insight into how to approach the problem of molecular identification, which could be useful to the research community.

**Weaknesses:**

- Some details still require further clarification and possibly some additional experiments. Please see questions below.

**Questions:**

- In equation 6, in $f$, can you clarify if the molecules in the candidate  set $\mathcal{C}_i$ correspond to the nodes in $\mathcal{G}$, or if $\mathcal{G}$ is a graph over candidate sets? Furthermore, In the problem definition, specifically in equation 6, what is the ranking over? Molecules in one candidate set $\mathcal{C}_i$? Or over all candidate sets $\mathcal{C}$?
- From section 3.2, is my understanding correct that you manually curated a novel dataset of molecules analyzed via tandem mass spectrometry?
- Since the co-occurrence graph is constructed using prior knowledge, have you ablated for the cases where the molecular co-occurrence graph has errors? For instance, what happens if some of the edges in the co-occurrence graph are removed or additional edges are added?
- In a similar vein, for the ablation in Table 2, what is the architecture used in the w/o graph setting?
- The method uses SMILES representations of molecules, which are encoded into an embedding space. Have the authors considered other molecular representations (such as molecular graphs)? How would performance of the method change when using different molecular representations?


Minor comments:

- "environmental co-occurrence" seems pertinent to this work and is mentioned several times in the abstract and introduction before a clear definition is provided. Possibly defining this term earlier would be beneficial to the reader.
- In section 5, it would be helpful to state a summary of the research questions in the title of each respective sub-section.

---

> ### Author Response · Authors · 2025-11-20
> **Response to Reviewer 9kL1**
>
> We sincerely appreciate your time and thoughtful comments on our work. For clarity, we provide point-by-point responses to your questions as follows:
>
> - **Q1:** Each node in $\mathcal{G}$ represents a compound, and nodes with unknown molecular structures are associated with a candidate set $\mathcal{C}_i$ containing possible molecular structures.
> The graph $\mathcal{G}$ is constructed based on spectral similarity and concentration correlations across environmental samples.
> For a compound $v_i$, the ranking in Equation (6) is performed within its own candidate set $\mathcal{C}_i$, rather than across all candidates $\mathcal{C}$.
> - **Q2:** Yes, that is correct. We collaborated with environmental scientists to curate in-situ sediment samples from the Taihu Lake in China, and we additionally processed three publicly available datasets related to human metabolism. Unlike prior tandem mass spectrometry datasets, our curated datasets also retain the concentration levels of substances across samples, which enables the model to exploit inter-compound correlations for molecular structure prediction.
> - **Q3:** Yes, we have examined the robustness of MoleRanker to potential errors in the molecular co-occurrence graph.
> Specifically, during graph construction, we varied the thresholds $\tau_s, \tau_c$ that determines the presence of edges in heterogeneous molecule co-occurrence graph to simulate cases where some edges are removed or added. As shown in Figure 4, the model’s performance remains stable within a reasonable threshold range, demonstrating that MoleRanker is not sensitive to moderate graph perturbations. However, when the threshold is set too low or too high, the resulting graph becomes excessively dense or sparse, respectively, which deteriorates performance due to distorted relational structure. In practice, we therefore recommend selecting an appropriate threshold using a small validation subset of nodes with known labels to maintain a balanced graph connectivity.
> - **Q4:** In the *w/o graph* setting, we remove the multiplex-relation message passing component on the constructed heterogeneous molecular co-occurrence graph. This ablation is used to verify the necessity and effectiveness of incorporating the heterogeneous molecular co-occurrence graph.
> - **Q5:** Our MoleRanker is agnostic to the specific SMILES encoder, and can in principle accommodate alternative molecular representations, including graph-based encoders. In the revised manuscript, we further evaluate MoleRanker with **MolT5** and **MolFormer** as alternative SMILES encoders to examine whether the framework remains effective under different molecular representations. Since MolT5 and MolFormer are pretrained on large-scale SMILES corpora and thus capture richer chemical semantics than traditional RDKit ECFP-4 fingerprints, MoleRanker equipped with these pretrained encoders achieves notably stronger performance. This confirms that our framework can effectively leverage improved molecular representations.
>
> For your convenience, we provide the experimental results below:
>
> | Methods | Sediments Hits@1| Sediments Hits@10| Sediments Hits@20| Sediments MRR| MTBLS146 Hits@1| MTBLS146 Hits@10| MTBLS146 Hits@20| MTBLS146 MRR| MTBLS265 Hits@1| MTBLS265 Hits@10| MTBLS265 Hits@20| MTBLS265 MRR↑ | MTBLS746 Hits@1| MTBLS746 Hits@10| MTBLS746 Hits@20| MTBLS746 MRR|
> |-|-|-|-|-|-|-|-|-|-|-|-|-|-|-|-|-|
> | MoleRanker(RDKit)         | 16.74 ± 0.93      | 33.02 ± 2.71       | 48.84 ± 2.08       | 0.226 ± 0.005  | 57.04 ± 1.81     | 73.33 ± 2.77      | 77.04 ± 1.48      | 0.604 ± 0.021 | 11.67 ± 2.19     | 23.67 ± 2.38      | 29.67 ± 1.45      | 0.157 ± 0.013 | 4.65 ± 0.00      | 19.07 ± 1.74      | 25.58 ± 0.00      | 0.095 ± 0.008 |
> | MoleRanker(MolT5)         | 27.91 ± 1.47      | 57.67 ± 6.31       | 69.30 ± 4.26| 0.365 ± 0.021  | 60.74 ± 10.63    | 86.67 ± 5.02      | 90.37 ± 4.44      | 0.696 ± 0.089 | 85.33 ± 1.46     | 93.87 ± 1.60      | 95.47 ± 1.81      | 0.883 ± 0.014 | 67.44 ± 2.94     | 81.86 ± 4.74      | 85.58 ± 2.28      | 0.731 ± 0.020 |
> | MoleRanker(MolFormer)     | 30.23 ± 0.00      | 59.53 ± 1.86       | 65.58 ± 2.28       | 0.397 ± 0.004  | 57.78 ± 5.02     | 71.85 ± 5.02      | 80.00 ± 2.96| 0.631 ± 0.045 | 75.47 ± 1.81     | 82.93 ± 0.53| 84.53 ± 1.07      | 0.783 ± 0.011 | 31.63 ± 3.15|52.09 ± 2.37|63.26 ± 4.97|0.396 ± 0.028|
>
> - **Q6:** *Environmental co-occurrence* refers to the phenomenon that compounds sharing similar functional groups often exhibit correlated concentration levels across environmental samples.
> This concept has been illustrated and described in Figure 1(c) of the paper.
> - **Q7:** We have refined the titles of all sub-sections in Section 5 to clearly reflect the main research questions addressed in each part, and we will update the revised manuscript accordingly as soon as possible.
>
> We once again thank you for your constructive and insightful feedback, and we hope that our responses address your concerns.

---

> > ### Comment · Reviewer_9kL1 · 2025-11-27
> >
> > Thank you for your detailed rebuttal! The authors have addressed my questions and concerns. Moreover, thank you for clarifying my question regarding the curation of a novel dataset, this is a valuable contribution. With this, I will maintain my positive rating of this paper, and increase my contribution score.
> >
> > However, it seems that all other reviewers raised questions and concerns regarding incorporating recent and competitive baselines. I lean to agree with the other reviewers on this point, and am willing to further raise my score if the authors address these concerns. I will monitor the discussion with other reviewers on this point.

---

> ### Author Response · Authors · 2025-12-02
>
> Thank you very much for your positive evaluation of our work and for your additional comments. Following the suggestions from other reviewers, we conduct extensive literature research and consult domain experts. Based on these investigations, we select **8 additional baselines** and perform fair and comprehensive experimental comparisons.
>
> - For heterogeneous graph neural networks, we include R-GCN [1], HAN [2], HetGNN [3], HGT [4], and SeHGNN [5].
> - In addition, following the MassSpecGym benchmark [6], we incorporate DeepSets, SMILES Transformer, and SELFIES Transformer, where the latter two are *de novo* generation methods.
>
> The experimental results show that MoleRanker continues to exhibit superior performance compared with other heterogeneous GNN models, indicating that our model architecture is better suited for capturing chemical constraints and environmental co-occurrence among molecules. Furthermore, the *de novo* generation methods behave as expected and perform extremely poorly due to the lack of large paired MS–SMILES data, yielding zero accuracy.
>
>
> |Methods| Sediments Hits@1|Sediments Hits@10| Sediments Hits@20| Sediments MRR| MTBLS146 Hits@1| MTBLS146 Hits@10| MTBLS146 Hits@20 | MTBLS146 MRR|
> | - | - | - | - | - | - | -| - | - |
> | R-GCN | 11.63 ± 1.47| 35.81 ± 2.37 | 45.12 ± 1.86| 0.1948 ± 0.0101| 53.33 ± 5.54| 70.37 ± 4.06| 73.33 ± 4.32 | 0.5769 ± 0.0497|
> | HAN| 11.16 ± 0.93| 33.02 ± 1.43| 43.26 ± 2.37| 0.1861 ± 0.0086| 52.59 ± 4.32| 68.15 ± 3.78| 72.59 ± 1.81| 0.5681 ± 0.0343|
> | HetGNN| 12.56 ± 2.37| 36.28 ± 2.37| 44.19 ± 2.94| 0.1989 ± 0.0195| 48.89 ± 3.63| 68.89 ± 3.78| 72.59 ± 5.02| 0.5578 ± 0.0251|
> | HGT| 12.56 ± 2.37| 34.88 ± 0.00| 43.26 ± 2.37| 0.1970 ± 0.0160| 50.37 ± 3.78| 72.59 ± 1.81|74.07 ± 2.34| 0.5627 ± 0.0258|
> |SeHGNN| 11.16 ± 0.93| 37.21 ± 4.88| 44.65 ± 2.71| 0.1930 ± 0.0107| 51.85 ± 4.06| 70.37 ± 3.31| 74.07 ± 2.34| 0.5700 ± 0.0316|
> | DeepSets| 3.70 ± 0.00| 22.22 ± 0.00| 31.48 ± 7.86 | 0.0703 ± 0.0017|0.00 ± 0.00  |0.67 ± 0.94|1.33 ± 0.00|0.0060 ± 0.0014|
> | MoleRanker | **16.74 ± 0.93** | 33.02 ± 2.71     | **48.84 ± 2.08** | **0.2257 ± 0.0052** | **57.04 ± 1.81** | **73.33 ± 2.77** | **77.04 ± 1.48** | **0.6044 ± 0.0214** |
>
> |Methods|MTBLS265 Hits@1|MTBLS265 Hits@10|MTBLS265 Hits@20|MTBLS265 MRR|MTBLS746 Hits@1|MTBLS746 Hits@10|MTBLS746 Hits@20|MTBLS746 MRR|
> | - | - | - | - | - | - | - | - | - |
> | R-GCN      | 8.80 ± 1.60            | 22.67 ± 1.46           | 29.07 ± 2.13           | 0.1416 ± 0.0134           | 3.26 ± 1.14           | 19.53 ± 1.14    | 25.58 ± 0.00           | 0.0838 ± 0.0030|
> | HAN        | 7.73 ± 1.00 | 22.67 ± 1.89| 29.07 ± 2.13           | 0.1300 ± 0.0064 | 4.19 ± 0.93           | 19.53 ± 1.14    | 25.12 ± 0.93 | 0.0903 ± 0.0068 |
> | HetGNN| 6.13 ± 0.65| 21.07 ± 2.29 | 26.40 ± 3.71| 0.1135 ± 0.0055| 4.65 ± 0.00           | 19.07 ± 1.74    | 25.58 ± 0.00           | 0.0936 ± 0.0090  |
> | HGT | 8.00 ± 0.84 | 22.67 ± 1.46 | 26.93 ± 2.44  | 0.1298 ± 0.0092 | 4.19 ± 0.93| 20.47 ± 0.93    | 25.58 ± 0.00 | 0.0906 ± 0.0053  |
> | SeHGNN     | 9.60 ± 1.00| 21.87 ± 2.75 | 27.47 ± 2.17  | 0.1398 ± 0.0145 | 3.26 ± 1.14  | 20.00 ± 2.37| 25.58 ± 0.00 | 0.0882 ± 0.0064|
> | DeepSets     | 2.33 ± 3.29| 5.81 ± 1.64   |5.81 ± 1.64| 0.0116 ± 0.0010 | 0.00 ± 0.00 | 2.33 ± 0.00 | 2.33 ± 0.00  |0.0065 ± 0.0007|
> | MoleRanker | **11.67 ± 2.19** | **23.67 ± 2.38** | **29.67 ± 1.45** | **0.1567 ± 0.0130** | **4.65 ± 0.00** | 19.07 ± 1.74|**25.58 ± 0.00**|**0.0949 ± 0.0083**|
>
> |Methods|Sediments Hits@1| Sediments Hits@10|Sediments Hits@20|MTBLS146 Hits@1|MTBLS146 Hits@10|MTBLS146 Hits@20|
> | - | - | - | -| - | - | - |
> | SMILES Transformer | 0.00 ± 0.00 | 0.00 ± 0.00     |0.00 ± 0.00 | 0.00 ± 0.00 |0.00 ± 0.00 | 0.00 ± 0.00 |
> | SELFIES Transformer | 0.00 ± 0.00 | 0.00 ± 0.00    |0.00 ± 0.00| 0.00 ± 0.00 | 0.00 ± 0.00| 0.00 ± 0.00 |
> | MoleRanker | **16.74 ± 0.93** | **33.02 ± 2.71**     | **48.84 ± 2.08** | **57.04 ± 1.81** | **73.33 ± 2.77** | **77.04 ± 1.48** |
>
> |Methods|MTBLS265 Hits@1| MTBLS265 Hits@10|MTBLS265 Hits@20|MTBLS746 Hits@1|MTBLS746 Hits@10|MTBLS746 Hits@20|
> | - | - | - | -| - | - | - |
> | SMILES Transformer | 0.00 ± 0.00 | 0.00 ± 0.00     |0.00 ± 0.00 | 0.00 ± 0.00 |0.00 ± 0.00 | 0.00 ± 0.00 |
> | SELFIES Transformer | 0.00 ± 0.00 | 0.00 ± 0.00    |0.00 ± 0.00| 0.00 ± 0.00 | 0.00 ± 0.00| 0.00 ± 0.00 |
> | MoleRanker | **11.67 ± 2.19** | **23.67 ± 2.38** | **29.67 ± 1.45** | **4.65 ± 0.00** | **19.07 ± 1.74**|**25.58 ± 0.00**|
>
>
> [1] Modeling Relational Data with Graph Convolutional Networks, European semantic web conference, 2018.
>
> [2] Heterogeneous Graph Attention Network, WWW, 2019.
>
> [3] Heterogeneous Graph Neural Network, KDD, 2019.
>
> [4] Heterogeneous Graph Transformer, WWW, 2020.
>
> [5] Simple and Efficient Heterogeneous Graph Neural Network, AAAI, 2023.
>
> [6] MassSpecGym: A benchmark for the discovery and identification of molecules, NeurIPS, 2024.

---

### Author Response · Authors · 2025-11-20
**General Response**

Dear Program Chairs, (Senior) Area Chairs and Reviewers,

We appreciate all reviewer's time and valuable comments. Below is a concise summary of the review.

## Reviewer Acknowledgement
1. **(Insight) Novel and interesting insight**
- Reviewer `9kL1`: *...offers a novel insight into how to approach the problem of molecular identification, which could be useful to the research community.*
- Reviewer `NbWy`: *The core idea is interesting and intuitive,...*
2. **(Dataset and Method) New dataset and novel method.**
- Reviewer `9kL1`: *The authors construct a new tandem mass spectrometry dataset...*; *The authors introduce a novel method for molecular identification,....*
- Reviewers `sthq`: *...it is a novel approach to include the correlation information from the concentration of molecules to improve the model's performance.*
- Reviewers `8Hq5`: *The integration of a dual-tower scoring mechanism with a pairwise Bayesian Personalized Ranking (BPR) objective exhibits practical value for large candidate sets and severe class imbalance problems.*
3. **(Experiments) Comprehensive empirical validation.**
- Reviewers `9kL1`: *...validate it through comprehensive empirical experiments.*
- Reviewers `8Hq5`: *The experiments conducted in this study are relatively comprehensive, and the results are favorable.*
- Reviewers `NbWy`: *...the reported metrics exhibit improvements, and the ablation studies substantiate the effectiveness of both information sources.*
- Reviewers `sthq`: *The proposed method outperforms several baselines,...*
4. **(Presentation) Clearly-written and well presentation.**
- Reviewers `9kL1`: *The paper is clearly written and well presented....*
- Reviewers `NbWy`: *I believe the paper is clearly written and conveys the authors’ ideas well.*
- Reviewers `8Hq5`: *The visualization outcomes are presented with commendable clarity.*

We sincerely appreciate the reviewers’ thoughtful engagement once again and believe that our responses address their concerns. Moreover, we would be glad to further discuss any aspect of the work with you.

Best regrads,

Authors

---

> ### Author Response · Authors · 2025-12-03
>
> # Summary of Common Concerns and Our Clarifications
>
> We sincerely thank all reviewers for their valuable questions and insightful comments.
> Across the reviews, we identify three core thematic concerns raised by multiple reviewers.
> Below, we summarize these concerns and clarify our perspectives.
> We also provide point-by-point responses under each reviewer’s discussion thread.
> In addition, we have updated the manuscript accordingly, with all newly added content highlighted in red.
>
> ---
>
> ## 1. Contribution of This Work as an AI4Science Paper
>
> Several reviewers raised questions regarding the contribution of our work. Our intention is clear: **MoleRanker is an AI for Science contribution** designed to serve two complementary segments of the ICLR readership:
>
> - **The science community**, who seek to understand how AI methods can address practical scientific challenges in molecular identification, overcome limitations of traditional analytical workflows, and push toward AI for Science.
> - **The AI/ML methodology community**, who require real scientific problems that can be properly formulated as machine learning tasks and need rigorous, scientifically grounded, and challenging benchmarks to drive new method or architecture designs.
>
> Accordingly, our contributions span three perspectives:
>
> 1. We formulate molecular identification as a molecular-structure ranking task and construct a benchmark consisting of four datasets across two scientific domains.
> 2. We translate two well-supported scientific observations into a heterogeneous molecular co-occurrence graph, providing a principled modeling framework instantiated by MoleRanker.
> 3. We conduct a complete, reproducible experimental benchmark enabling rigorous validation of new models.
>
> Throughout the paper, we carefully balance scientific and methodological presentation. We aim to ensure that readers understand both the process of scientific modeling and the underlying domain principles.
>
> ---
>
> ## 2. Choice of Baselines
>
> Reviewers asked why we selected the particular baselines included in the paper. Our choices are both practical and comprehensive:
>
> - **For the science community**, we consulted domain experts and reviewed state-of-the-art works published in *Nature*, *Science*, and related venues. Based on this survey, we selected the three most widely used and representative baselines (MetFrag, CFM-ID, and SIRIUS).
> - **For the AI/ML community**, we incorporated classical machine learning models and graph neural networks. These highlight that the task can indeed be formulated as a spectrum-driven molecular structure ranking problem and that our heterogeneous molecular co-occurrence graph faithfully captures *chemical constraints* and *environmental co-occurrence* patterns.
>
> Following reviewer suggestions, we further added additional baselines, including models targeting heterogeneous relations and several *de novo* generation approaches.
>
> We clarify that our baseline selection is driven by two principles:
> (1) practical relevance to real scientific workflows, and
> (2) scientific value for promoting progress in both communities.
>
> ---
>
> ## 3. Requests for More Baselines and Efficiency Experiments
>
> Reviewers asked for additional baselines and runtime comparisons. We have included the following experiments while ensuring fair evaluation:
>
> - **We added 8 additional methods, including 5 heterogeneous GNN models, 2 *de novo* generation methods, and 1 molecular structure ranking methods**. Experimental results demonstrate that *de novo* generation methods remain extremely challenging due to their dependence on large amounts of high-quality paired MS/MS–SMILES data. Meanwhile, MoleRanker’s multiplex-relation message passing effectively models both chemical constraints and environmental co-occurrence, achieving state-of-the-art performance. We also believe that this pioneering work will inspire the AI community to explore stronger architectures for molecular ranking tasks.
> - **We included training and inference runtime for all methods across all four datasets.** The results demonstrate that replacing traditional domain-specific pipelines with ML methods dramatically improves computational efficiency, highlighting the significant advantages of this AI for Science direction.
>
> ---

---

### Comment · Area_Chair_WpBZ · 2025-11-27

Thank you very much for the reviewer's comments and the author's positive response. As there is not much time left for discussion, please actively participate in the discussion and provide a more valuable response to this paper.

---

### Meta-Review · Area_Chair_KpVR · 2025-12-14

**Summary:**

This paper proposes MoleRanker, a heterogeneous graph neural network for spectrum-driven molecular structure ranking that integrates chemical constraints with environmental co-occurrence information. The problem is well motivated, the datasets are valuable, and several reviewers acknowledged the potential practical impact of the approach.

However, fundamental concerns remain unresolved. Multiple reviewers questioned the realism of the problem formulation, particularly the assumption that a randomly selected subset of nodes in a co-occurrence graph has reliable molecular labels, and raised concerns that graph-based label propagation may limit generalization to new datasets or applications. In addition, reviewers expressed persistent doubts about the methodological novelty, viewing the approach largely as an engineering integration of existing GNN components and heterogeneous relations. Despite clarifications and additional experiments in the rebuttal, concerns regarding generalization, baseline coverage, and scalability were only partially addressed, and at least one reviewer remained unconvinced.

Given these remaining issues and the mixed reviewer consensus, I recommend rejection at this time.

**Reviewer Concerns:**

Reviewer 9kL1 raised several clarification-oriented questions regarding the problem formulation and experimental setup, including the interpretation of Equation 6 and the ranking objective, dataset construction from tandem mass spectrometry, robustness to errors in the co-occurrence graph, architectural details of the ablation settings, and the choice of molecular representations. Minor comments concerned the early definition of “environmental co-occurrence” and clearer structuring of experimental sections.

The authors provided detailed and precise responses, clarifying the formulation, experimental design, and modeling choices, and addressing robustness and representation-related questions. These explanations effectively resolved the reviewer’s concerns, and the reviewer expressed a clearly positive attitude following the rebuttal.

---

Reviewer sthq questioned the realism and practical applicability of the problem setup, arguing that assuming a random subset of graph nodes has known molecular labels is unrealistic in real-world mass spectrometry scenarios and may lead to biased training. The reviewer further expressed concern that graph-based label propagation limits generalizability, as models trained on a specific co-occurrence graph may not transfer to new applications or networks. Additional criticisms included the perception that the GNN-based design offers limited methodological novelty and that the experimental comparison omits more recent deep learning–based MS/MS identification methods.

Although the authors provided detailed responses and clarifications, the reviewer remained unconvinced, emphasizing that their main concern lies in the assumption of randomly labeled graph nodes and the resulting loss of generalization once a graph structure is imposed. The reviewer viewed the rebuttal as largely addressing the non-graph setting and raised new concerns regarding the applicability of graph-based learning in realistic deployment scenarios, leaving their core objections unresolved.

---

Reviewer 8Hq5 raised concerns regarding the omission of several recent and directly relevant baselines, particularly methods based on topological or spectral graph modeling and multi-relational molecular graphs. The reviewer also noted insufficient discussion of practical limitations and generalization, including potential distributional shifts, batch variability, and candidate quality differences. Additional concerns involved the lack of justification for excluding other advanced heterogeneous or relational GNN variants, as well as the absence of runtime and scalability analysis despite claims of improved computational efficiency.

In the rebuttal, the authors added additional experiments with different GNN variants, provided computational cost and scalability details, and clarified several points that were previously misunderstood. These responses substantially improved the evaluation and largely addressed the reviewer’s concerns.

----
Reviewer NbWy questioned the novelty of the proposed heterogeneous co-occurrence graph, viewing it primarily as an engineering combination of additional information sources rather than a deeper methodological advance. They also raised concerns about insufficient experimental specification, including unclear baseline configurations (e.g., whether homogeneous GNN baselines were trained on identical relation layers), an under-specified training and evaluation pipeline, and the absence of computational efficiency results despite claims of lower cost. Additional questions were raised regarding the strength of the chosen baselines and the near-zero performance of traditional methods such as MetFrag, CFM-ID, and SIRIUS on several datasets.

In the rebuttal, the authors clarified the core contributions, provided detailed explanations of the baseline setups and training pipeline, and added discussion and evidence regarding computational efficiency. They also explained the observed performance of traditional methods in the given experimental setting. These clarifications addressed most of the reviewer’s concerns and improved the transparency of the experimental evaluation.

**Reviewer Scores:**

Reviewer 9kL1 expressed a positive attitude following the rebuttal and may further increase their score. Reviewer sthq remains strict and raised additional concerns regarding the model’s practical applicability and generalization, making a score increase unlikely. Reviewer 8Hq5’s concerns were largely resolved, and they may revise their score upward and provide more positive feedback. Reviewer NbWy is likely to maintain their original score, as their primary concerns relate to novelty problem.

---

### Decision · Program_Chairs · 2026-01-26

Reject